# SINGLE-SAMPLE TEST-TIME REINFORCEMENT LEARNING FOR VISION-LANGUAGE MODELS

## ABSTRACT

While Test-Time Reinforcement Learning (TTRL) has shown promise for adapting language models without ground truth answers, its application to vision-language tasks remains unexplored. Similarly, existing TTRL methods require multiple samples or known answers for optimization, limiting their practical applicability. We introduce Vision Reasoning Test-Time Reinforcement Learning (VR-TTRL), to our knowledge, the first framework to apply TTRL to vision-language models for visual reasoning tasks, enabling adaptation from a single unlabeled sample without any ground truth answers. Our approach leverages majority voting across model rollouts to generate pseudo-labels for self-supervision, combining the structured reasoning capabilities of vision-language models with the adaptive power of test-time reinforcement learning. Through experiments on segmentation and counting tasks, we demonstrate that VR-TTRL enables effective model adaptation using only a single unlabeled sample, achieving performance improvements over state-of-the-art baselines. This work suggests promising directions for further improving vision task performance through self-supervised adaptation and enabling models to better leverage their pre-trained capabilities during inference.

## 1 INTRODUCTION

Recent advancements in Vision-Language Models (VLMs) have marked a paradigm shift in their capacity for complex, multi-modal reasoning. The integration of reinforcement learning (RL) techniques, particularly following the introduction of algorithms like Group Relative Policy Optimization (GRPO) (Guo et al., 2025), has become a cornerstone for enhancing these models' cognitive abilities. Frameworks such as VisionReasoner (Liu et al., 2025b) and Seg-Zero (Liu et al., 2025a) have demonstrated that by applying carefully designed rewards, VLMs can be trained to generate an internal dialogue or Chain-of-Thought (CoT), significantly boosting their performance on intricate perception tasks like semantic segmentation and object counting. This process of cognitive reinforcement not only improves accuracy but also endows the models with a more transparent and interpretable reasoning process.

Despite these successes, a fundamental limitation persists: once trained, these powerful models are typically deployed as static artifacts. They struggle to adapt when faced with data from novel distributions or challenging, out-of-distribution examples encountered during inference. This performance degradation at test-time highlights a critical gap between pre-training and real-world application, where the diversity of data often exceeds what was seen during the training phase. The ability to dynamically adjust to new samples is therefore essential for robust and reliable deployment.

To bridge this gap, the paradigm of Test-Time Adaptation (TTA) has emerged as a promising solution (Sun et al. (2020), Akyurek et al. (2024), Behrouz et al. (2024)). TTA enables a model to adapt itself to test samples before making predictions. While early methods focused on Test-Time Training (TTT) with self-supervised losses, recent work has explored single-sample adaptation for domain generalization (Xiao et al., 2022), and recent advancements have introduced Test-Time Reinforcement Learning (TTRL) (Zuo et al., 2025). TTRL leverages techniques like self-consistency or majority voting across multiple stochastic rollouts to generate a pseudo-label or a reward signal from the test sample itself. However, existing approaches face key limitations. TTRL methods have so far been applied only to language models and rely on multiple test samples for majority-vote pseudo-labeling Zuo et al. (2025). Vision-based test-time adaptation methods require ground-truth labels during meta-

training Xiao et al. (2022), and recent single-sample RL work for visual reasoning Shao et al. (2025) performs offline training with labeled data rather than test-time adaptation. This leaves unexplored the combination of single-sample, label-free reinforcement learning adaptation for vision-language reasoning tasks.

We address these considerations by introducing VR-TTRL , the first framework to apply TTRL to vision-language models for visual reasoning tasks. Our key innovations are that we demonstrate that TTRL can be effectively applied to vision-language models, opening new possibilities for adaptive visual reasoning, and we show that effective adaptation can be achieved using only a single unlabeled sample without any ground truth answers, making the approach practically viable for real-world deployment. VR-TTRL similar to TTRL Zuo et al. (2025) leverages majority voting across model rollouts to generate reliable pseudo-labels, enabling the model to refine its reasoning process for each specific test sample.

Our contributions are as follows:

- We introduce, to our knowledge, the first application of Test-Time Reinforcement Learning (TTRL) to vision-language models, extending TTRL from language-only mathematical reasoning to visual reasoning tasks through a novel adaptation of the majority voting mechanism for vision-language model rollouts.
- We propose a single-sample optimization framework that enables TTRL adaptation using only one unlabeled test sample without ground truth answers, contrasting with existing TTRL methods that require batch processing of multiple samples or entire datasets.
- We develop Vision Reasoning Test-Time Reinforcement Learning (VR-TTRL), a framework that integrates vision-language reasoning capabilities with TTRL's self-supervision mechanism, enabling models to refine their visual reasoning process through majority voting across multiple reasoning trajectories generated for individual test samples.

## 2 RELATED WORKS

### 2.1 LARGE VISION-LANGUAGE MODELS FOR REASONING

The landscape of artificial intelligence has been reshaped by the advent of Large Vision-Language Models (VLMs), which integrate pre-trained vision encoders and large language models (LLMs) to perform sophisticated multi-modal tasks. Foundational architectures like LLaVA (Liu et al., 2023b) demonstrated the viability of this approach by connecting a vision encoder to an LLM using a simple projection layer, enabling instruction-following on visual inputs. Subsequent models, such as GPT-4V (Yang et al., 2023), have pushed the boundaries of zero-shot reasoning capabilities, establishing a high bar for visual understanding and interaction.

The field has since witnessed rapid development with various model families emerging. The Qwen family of models (Bai et al., 2023) has gained significant attention, with Qwen2.5-VL (Bai et al., 2025) representing a state-of-the-art vision-language model that demonstrates strong performance across diverse visual reasoning tasks. Other notable VLMs include LLaVA-1.5 (Liu et al., 2023a), which improved upon the original LLaVA architecture, and multimodal variants of popular language models such as Claude-3 (Anthropic, 2024) and Gemini (Google DeepMind, 2023). These models have established benchmarks for visual question answering, image captioning, and complex visual reasoning tasks.

More recently, research has shifted towards enhancing these models' cognitive and perceptual abilities through fine-tuning on specialized tasks. A prominent example is the VisionReasoner framework (Liu et al., 2025b), which adapts a base VLM using a multi-faceted reward function within a reinforcement learning paradigm. This reward function is designed to elicit complex reasoning by rewarding not only the final perceptual output, such as segmentation masks, bounding boxes, and keypoints; but also the generation of an explicit Chain-of-Thought (CoT) that verbalizes the model's internal reasoning process. This methodology has proven highly effective in improving performance on fine-grained visual tasks. Furthermore, this work aligns with a broader trend of creating modular systems where a reasoning VLM can be composed with specialized models, such as the Segment Anything Model (SAM) (Kirillov et al., 2023), to decouple high-level reasoning from high-fidelity pixel-level execution.

## 2.2 Reinforcement Learning for Large Models

Reinforcement Learning from Human Feedback (RLHF) has been instrumental in aligning LLMs with human intent, but its application to the vision domain introduces unique challenges. The DeepSeek-R1 model (Guo et al., 2025) represents a significant advancement in reasoning capabilities through reinforcement learning, demonstrating how iterative refinement can improve model performance on complex reasoning tasks. However, traditional RLHF approaches require extensive human feedback, which is costly and difficult to scale.

Recent work has explored self-supervised reinforcement learning paradigms that reduce reliance on human annotations. The TTRL (Test-Time Reinforcement Learning) framework (Zuo et al., 2025) introduced the concept of adapting models at inference time using their own predictions as pseudo-labels. This approach is particularly relevant to our work, as it addresses the fundamental challenge of model adaptation without external supervision. However, existing TTRL methods have primarily focused on text-only tasks, leaving a gap in the vision-language domain.

Group Relative Policy Optimization (GRPO) from Guo et al. (2025) represents another important development in reinforcement learning for large models. Unlike traditional Proximal Policy Optimization (PPO), GRPO operates on group-based advantage computation, making it particularly suitable for scenarios where relative performance within groups is more important than absolute advantage estimates. This approach aligns well with our test-time adaptation setting, where we seek to improve model performance relative to its initial predictions on a given sample.

## 2.3 Test-Time Adaptation and Domain Adaptation

Test-time adaptation has emerged as a crucial paradigm for addressing distribution shift in machine learning models. Traditional approaches often require access to labeled data from the target domain or assume the availability of multiple samples for adaptation. Recent work has explored various strategies, from gradient-based updates (Wang et al., 2021a) to entropy minimization (Wang et al., 2021b), and single-sample adaptation for domain generalization (Xiao et al., 2022). However, these methods have primarily focused on traditional computer vision tasks and have not been applied to vision-language reasoning tasks. Our approach extends this paradigm to vision-language models, enabling adaptation from a single sample using the model's own predictions as supervision signals through reinforcement learning.

# 3 Methods

Our approach, which we term VR-TTRL , adapts a pre-trained Vision-Language Model (VLM) to a single, unlabeled test sample using online reinforcement learning. We base our method on the core principles of Test-Time Reinforcement Learning (TTRL), where the key idea is to generate a reliable teaching signal from the model's own stochastic outputs. However, a significant challenge arises in complex vision-reasoning tasks where outputs are not simple strings but structured data, such as JSON containing coordinates and masks. In these scenarios, a standard majority vote based on exact string matching is ineffective.

We leverage the sophisticated reward function from the VisionReasoner framework, originally used to assess a prediction against a ground truth, to now measure the similarity between two different model-generated answers. The answer that exhibits the highest cumulative similarity to all other answers generated in a set of stochastic rollouts is chosen as the pseudo-label for adaptation. An illustration of our approach is shown in Fig. 1.

## 3.1 The VR-TTRL Adaptation Framework

Given a query, the model generates a rollout of candidate answers. We compute pairwise IoU and point-wise L1 distances among all candidates (see Section 3.2.2), aggregate these metrics into a consensus score and select the highest-scoring candidate as the pseudo-label.

For each candidate in the rollout, we compute a reward signal that consists of two components: (1) an accuracy score that measures alignment with the majority answer, and (2) a reasoning-format score that ensures the reasoning process. The policy optimization step updates the model parameters using

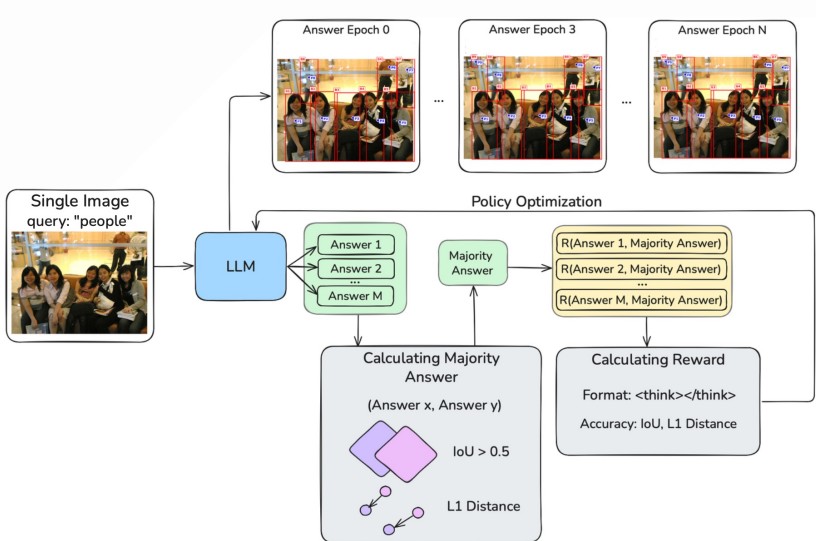

Figure 1: VR-TTRL Framework Overview. Given a single test image, the vision-language model generates multiple candidate answers through stochastic rollouts. The majority answer is established by computing pairwise IoU and L1 distance metrics between generated bounding boxes and points across all candidates. The candidate that maximizes the overall consensus score with other responses is selected as the pseudo-label. This majority answer is then used to compute reward signals for each candidate, combining accuracy-based rewards (measuring alignment with the majority answer) and reasoning-format rewards (ensuring the reasoning chains) to guide policy optimization.

these computed rewards, and the entire procedure iterates with new rollouts in subsequent epochs, enabling progressive refinement of the model's predictions.

## 3.2 REWARD FUNCTIONS

Our reward functions consist of two primary components: format rewards and accuracy rewards. Following the VisionReasoner framework (Liu et al., 2025b), we use target object bounding boxes and center points to calculate rewards rather than binary masks, as this approach provides better training efficiency and more precise geometric supervision. The accuracy rewards are calculated between all rollout answers, and the answer that maximizes the total reward among them is chosen as the majority answer. The GRPO algorithm uses these pre-calculated rewards for the majority answer and adds the format rewards on top to guide the optimization process.

### 3.2.1 FORMAT REWARDS

**Thinking Format Reward.** This reward constrains the model to output an explicit reasoning process between `<think>` and `</think>` tags, followed by the final answer between `<answer>` and `</answer>` tags. This format ensures that the model provides transparent reasoning while maintaining structured output.

**Answer Format Reward.** We use bounding boxes $\mathbf{B}_i$ and points $\mathbf{P}_i$ as the structured answer format, as this approach offers superior training efficiency compared to segmentation masks. This reward restricts the model to output answers in the format:

$$['\texttt{bbox\_2d}' : [x_1, y_1, x_2, y_2], '\texttt{point\_2d}' : [x_1, y_1], \ldots] \tag{1}$$

**Non-Repeat Reward.** To avoid repetitive patterns in the reasoning process, we split the thinking process into individual sentences and prioritize those with unique or non-repetitive reasoning patterns.

Table 1: Number of samples assessed in each benchmark dataset split.

| Type | Dataset | Split | # of samples |
|---|---|---|---|
| Segmentation | RefCOCO | testA | 200 |
| | RefCOCO+ | testA | 200 |
| | RefCOCOg | test | 200 |
| | ReasonSeg | test | 200 |
| | | val | 200 |
| Counting | Pixmo-Count | test | 200 |
| | | val | 200 |
| | CountBench | test | 200 |

### 3.2.2 ACCURACY REWARDS

**Bounding Box IoU Reward.** Given a set of $N$ predicted bounding boxes from one rollout answer and $K$ predicted bounding boxes from another rollout answer, this reward computes their Intersection-over-Union (IoU) scores. For each IoU exceeding 0.5, we increment the reward by $\frac{1}{\max\{N,K\}}$.

**Bounding Box L1 Reward.** Given a set of $N$ predicted bounding boxes from one rollout answer and $K$ predicted bounding boxes from another rollout answer, this reward computes their L1 distance. For each L1 distance below the threshold of 10 pixels, we increment the reward by $\frac{1}{\max\{N,K\}}$.

**Point L1 Reward.** Given a set of $N$ predicted points from one rollout answer and $K$ predicted points from another rollout answer, this reward computes matched L1 distance. For each L1 distance below the threshold of 30 pixels, we increment the reward by $\frac{1}{\max\{N,K\}}$.

The combination of these rewards ensures that the model learns to produce both well-structured outputs and accurate geometric predictions relative to the pseudo-label, while maintaining transparent reasoning processes throughout the adaptation procedure.

## 4 EXPERIMENTS

### 4.1 BENCHMARKS

We use six benchmarks to evaluate model performance across general vision perception tasks. Our evaluation includes fundamental tasks such as segmentation and counting. Specifically, we employ COCO Lin et al. (2014), RefCOCO(+/g) Kazemzadeh et al. (2014); Yu et al. (2016) and ReasonSeg Lai et al. (2024) for segmentation evaluation; PixMo-Count Deitke et al. (2024) and CountBench Paiss et al. (2023) for counting evaluation. The number of samples per benchmark is presented in Tab. 1, where the sample sizes were selected to provide comprehensive coverage of structured vision-language tasks while remaining computationally tractable for per-sample VR-TTRL adaptation, which operates on individual samples without requiring separate training or validation sets. Detailed descriptions of the evaluation metrics used for each benchmark (gIoU, cIoU and counting accuracy) are provided in the Appendix A.

### 4.2 EXPERIMENTAL SETUP

We test our VR-TTRL framework on the VisionReasoner-7B model, which is based on Qwen2.5-VL-7B-Instruct, and we also evaluate Qwen2.5-VL-7B-Instruct. The experimental setup follows a test-time reinforcement learning paradigm using Group Relative Policy Optimization (GRPO) as the advantage estimator.

**Training Parameters.** The learning rate is set to $5 \times 10^{-7}$ with a weight decay of $1 \times 10^{-2}$. We use a KL divergence coefficient of $1 \times 10^{-2}$ with low-variance KL loss to maintain policy stability. The maximum gradient norm is clipped at 1.0, and we employ parameter and optimizer offloading to manage memory constraints.

**Rollout Configuration.** Each adaptation episode generates 8 rollout candidates (if not specified differently in experiments) per sample with a temperature of 0.6. The rollout process uses 4 GPUs

with a memory utilization of 60% and disables chunked prefill for consistency. We limit the evaluation to 200 images per dataset to balance computational efficiency with comprehensive evaluation.

**Infrastructure.** All experiments are conducted on H100 GPUs with 32 CPU cores per task, running for up to 12 hours per split of evaluated dataset.

### 4.3 BASELINES

**Simple TTA via self-consistency** (no parameter update). For each test sample, we generate multiple stochastic rollouts with the base VLM and apply the same consensus mechanism as in VR-TTRL (pairwise IoU and L1-based similarity) to select a majority answer, but we do not perform any parameter updates. This corresponds to a self-consistency / majority-voting TTA baseline that only exploits sampling and ensembling at test time.

Additional baselines including **Tent-style entropy minimization**, **MEMO-style consistency TTA**, **SFT with single greedy pseudo-label**, and **SFT with majority pseudo-label** are evaluated and discussed in Appendix C.

### 4.4 MAIN RESULTS

Table 2 presents a comprehensive evaluation of our VR-TTRL framework across segmentation and counting tasks, comparing base models (Qwen2.5-VL-7B and VisionReasoner-7B) with their VR-TTRL -enhanced versions and the vote-only consensus baseline. Our results demonstrate that VR-TTRL provides consistent improvements across vision-language tasks, with effectiveness varying based on the underlying model's capabilities and task complexity.

**Segmentation tasks, VR-TTRL shows substantial improvements across all datasets**, with Qwen2.5-VL-7B achieving the most significant gains. The base Qwen2.5-VL-7B model shows remarkable improvements: +8.66% on ReasonSeg validation, +5.26% on ReasonSeg test, +9.91% on RefCOCO testA, +9.55% on RefCOCO+ testA, and +5.92% on RefCOCOg test, resulting in an overall improvement from 47.48% to 55.34% average performance. The VisionReasoner-7B model shows more modest but consistent improvements, reaching 71.17% average performance. The larger improvements observed for Qwen2.5-VL-7B suggest that VR-TTRL is particularly effective for models that start from a lower baseline, enabling rapid enhancement of reasoning capabilities that were not previously fine-tuned for these specific visual reasoning tasks.

**The counting results show consistent improvements across both models, demonstrating VR-TTRL effectiveness on this challenging task.** For Qwen2.5-VL-7B, VR-TTRL achieves substantial improvements: +7.50% on PixMo validation, +7.00% on PixMo test, and +4.00% on CountBench, advancing from 41.33% to 47.50% average performance—a +6.17% overall gain. VisionReasoner-7B demonstrates even stronger improvements than on segmentation task: +3.50% on PixMo validation, +2.50% on PixMo test, and +2.50% on CountBench, improving from 77.50% to 80.50% average performance. These results indicate that VR-TTRL effectively handles both geometric reasoning (segmentation) and enumeration tasks (counting).

**The vote-only consensus baseline demonstrates that RL updates provide substantial value beyond pure selection.** For Qwen2.5-VL-7B on segmentation, the consensus-only baseline underperforms the base model (44.38% vs. 47.48%), while VR-TTRL achieves 55.34%, representing a +10.96% gain over consensus alone. This demonstrates that the RL adaptation is essential—simply selecting consensus predictions without model updates is insufficient and can even be detrimental. For VisionReasoner-7B, consensus alone maintains similar performance to the base model (69.98% vs. 70.15%), while VR-TTRL provides meaningful improvement to 71.17%, confirming that the gains come from RL-driven model adaptation rather than selection artifacts.

**The overall results demonstrate two distinct patterns of improvement that highlight the versatility of our approach.** For VisionReasoner-7B, which was specifically designed for structured vision-language tasks, VR-TTRL provides stable improvements through refinement of existing capabilities, achieving 71.17% on segmentation and 80.50% on counting respectively. This demonstrates that the pseudo-labeling mechanism works effectively for models already optimized for the target tasks. For the base Qwen2.5-VL-7B model, VR-TTRL enables rapid enhancement of capabilities that were not previously fine-tuned, achieving substantial improvements of +7.86% on segmentation

Table 2: Performance comparison on segmentation tasks and counting tasks. We use SAM2 for vision-language models if necessary in segmentation tasks. There are 5 epochs per image. For segmentation tasks we check gIoU (average of per image IoU); for counting we match the number of returned bounding boxes with ground truth. The results demonstrate two distinct patterns of improvement: VisionReasoner-7B shows stable improvements through refinement of existing capabilities, while Qwen2.5-VL-7B achieves rapid enhancement of previously untuned capabilities, using only single samples without ground truth supervision.

| Method | Segmentation | | | | Avg. | Counting | | | Avg. |
|---|---|---|---|---|---|---|---|---|---|
| | ReasonSeg | RCO | RCO+ | RCOg | | Pixmo | | Count | |
| | val | test | testA | testA | test | | val | test | test | |
| Task-specific Models | | | | | | | | | | |
| Qwen2.5-VL-7B | 41.89 | 38.95 | 54.90 | 51.20 | 50.46 | 47.48 | 46.50 | 44.00 | 33.50 | 41.33 |
| w/ Rollouts + Consensus | 39.36 | 38.24 | 49.95 | 47.46 | 46.91 | 44.38 | 53.50 | 32.50 | 32.50 | 39.50 |
| w/ VR-TTRL | **50.55** | **44.21** | **64.81** | **60.75** | **56.38** | **55.34** | **54.00** | **51.00** | **37.50** | **47.50** |
| VisionReasoner-7B | 66.08 | 59.65 | 77.27 | 76.08 | 71.66 | 70.15 | 78.50 | 69.00 | 85.50 | 77.50 |
| w/ Rollouts + Consensus | 63.34 | 60.09 | 76.45 | **77.24** | 72.81 | 69.98 | 78.50 | 66.50 | 85.50 | 76.83 |
| w/ VR-TTRL | **66.28** | **62.59** | **77.60** | 76.19 | **73.19** | **71.17** | **82.00** | **71.50** | **88.00** | **80.50** |

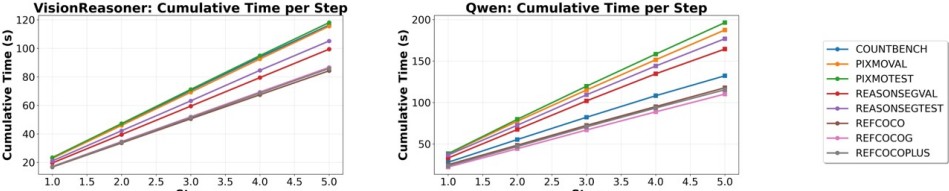

Figure 2: Computational timing analysis showing linear scaling of adaptation time across VR-TTRL steps. VisionReasoner demonstrates faster and more consistent timing (16-23s per step) compared to Qwen (22-41s per step), with cumulative time increasing predictably across all datasets.

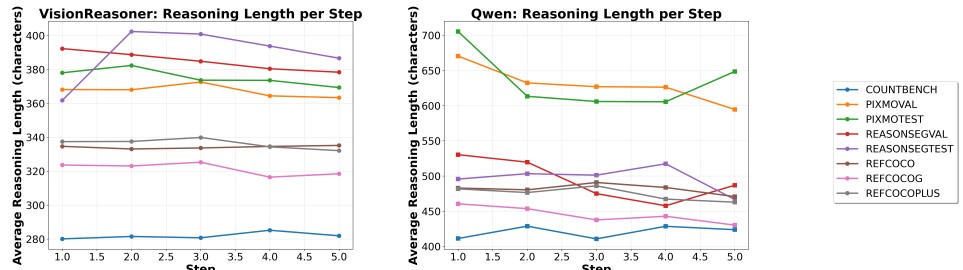

Figure 3: Reasoning length analysis showing character count in thinking chains across VR-TTRL steps. Models generally maintain or slightly reduce reasoning length during adaptation, with Vision-Reasoner showing more stable patterns and Qwen exhibiting greater variability and adaptation-driven changes.

and +6.17% on counting without any ground truth supervision, using only a single sample at a time. Crucially, the consistent outperformance of VR-TTRL over the vote-only baseline isolates the contribution of RL updates, confirming that test-time reinforcement learning provides genuine adaptation benefits beyond simple consensus-based selection.

**Our computational analysis reveals that VR-TTRL exhibits predictable scaling and refined reasoning patterns during adaptation.** The timing analysis (see Fig. 2) demonstrates linear scaling with step count, making computational budget estimation feasible for deployment. The reasoning length analysis (see Fig. 3) shows that models become more concise rather than deeper in their thinking, with VisionReasoner maintaining consistent patterns (280-401 characters) while Qwen shows greater variability (413-700 characters) and more dramatic adaptation changes.

Table 3: Generation vs. update ablation study and anti-consensus validation. We compare standard VR-TTRL (5 generation steps, 5 updates) with Single Gen variant (1 generation step, 10 updates), and include VR-TTRL (-) (5 generation steps, 5 updates) which uses the negative of consensus (lowest similarity rollout) as pseudo-label while maintaining the same format reward to validate that consensus quality matters. Single Gen achieves comparable or superior performance, demonstrating that extended updates are more effective than generation diversity. The poor performance of VR-TTRL (-) confirms that consensus selection is critical—not any reward signal suffices.

| Method | ReasonSeg val | RefCOCOg testA |
|---|---|---|
| Qwen2.5-VL-7B | 41.89 | 50.46 |
| Qwen2.5-VL-7B w/ VR-TTRL (-) | 49.47 | 53.71 |
| Qwen2.5-VL-7B w/ VR-TTRL | 50.55 | 56.38 |
| Qwen2.5-VL-7B w/ VR-TTRL (Single Gen) | **51.80** | **57.05** |
| VisionReasoner-7B | 66.08 | 71.66 |
| VisionReasoner-7B w/ VR-TTRL (-) | 60.68 | 69.87 |
| VisionReasoner-7B w/ VR-TTRL | **66.28** | **73.19** |
| VisionReasoner-7B w/ VR-TTRL (Single Gen) | 66.20 | 73.13 |

## 4.5 ABLATION STUDY

**Our analysis reveals that adaptation effectiveness depends more on update iterations than generation diversity**, challenging conventional assumptions about consensus building. Table 3 shows that a "Single Gen" variant (1 generation step, 10 updates) achieves comparable or superior performance to the standard approach (5 generation steps, 5 updates). For Qwen2.5-VL-7B, Single Gen outperforms standard on ReasonSeg validation (+1.25%, 50.55% to 51.80%) and RefCOCOg testA (+0.67%, 56.38% to 57.05%), while VisionReasoner-7B maintains nearly identical performance across both variants. This finding indicates that iterative self-supervision through extended updates is more critical than multiple candidate generations, suggesting that significant improvements can be achieved with reduced computational overhead for resource-constrained deployment.

**To validate that consensus quality matters rather than any arbitrary reward signal being sufficient**, we conduct an anti-consensus experiment (VR-TTRL (-)) where we deliberately select the negative of consensus—the rollout with lowest IoU/L1 similarity to other predictions—as the pseudo-label, while keeping the format reward identical to isolate the effect of consensus selection. The results in Table 3 show that VR-TTRL (-) significantly underperforms both the base model and standard VR-TTRL . For Qwen2.5-VL-7B, VR-TTRL (-) achieves only 49.47% on ReasonSeg validation and 53.71% on RefCOCOg testA, compared to 50.55% and 56.38% for standard VR-TTRL —demonstrating that optimizing toward poor pseudo-labels degrades performance despite maintaining proper format rewards. For VisionReasoner-7B, VR-TTRL (-) drops to 60.68% and 69.87%, substantially below both the base model (66.08%, 71.66%) and standard VR-TTRL (66.28%, 73.19%). This validates that our consensus mechanism provides meaningful signal: selecting high-quality pseudo-labels through majority voting is essential, and not just any reward signal will improve the model. These results directly address concerns about spurious rewards, confirming that VR-TTRL ' gains stem from principled consensus selection rather than arbitrary RL optimization.

**Our ablation studies reveal optimal hyperparameter settings that balance performance gains with computational efficiency.** Table 4 shows that 10 updates per image provide the best performance (5.52% gIoU, 8.68% cIoU improvement) before diminishing returns set in, while computational cost scales linearly from 20.81 to 306.86 seconds per image. Table 5 demonstrates that rollout size 8 is sufficient for effective consensus building, with minimal gains from larger rollouts (2.14% to 2.19% gIoU) but significant computational overhead (86.7 to 229.8 seconds per image). Unscaled ablation results are provided in the Appendix B. These findings indicate that VR-TTRL achieves substantial improvements with modest computational requirements, making it practical for real-world deployment.

Table 4: Effect of #updates per image on Reason-Seg test using VisionReasoner (rollout size 8). Performance peaks at 10 updates before diminishing returns, demonstrating optimal adaptation without overfitting.

| #Updates | gIoU% ↑ | cIoU% ↑ | Time (s/img)↓ |
|---|---|---|---|
| 1 | 0 | 0 | **20.81** |
| 2 | 0.32 | -0.65 | 41.86 |
| 3 | 3.86 | 6.46 | 62.73 |
| 5 | 4.93 | 7.49 | 104.53 |
| 10 | **5.52** | **8.68** | 206.42 |
| 15 | 4.68 | 7.19 | 306.86 |

Table 5: Rollout size analysis on RefCOCOg using VisionReasoner. Increasing rollout size from 8 to 32 provides minimal performance gains but triples computational cost, indicating that small rollouts are sufficient for effective consensus.

| Rollout | gIoU%↑ | cIoU%↑ | Time (s/img)↓ |
|---|---|---|---|
| 8 | 2.14 | 1.13 | **86.7** |
| 16 | 2.01 | 0.38 | 132.4 |
| 32 | **2.19** | **1.18** | 229.8 |

Q: A purple suitcase, being pulled by a woman with blonde hair

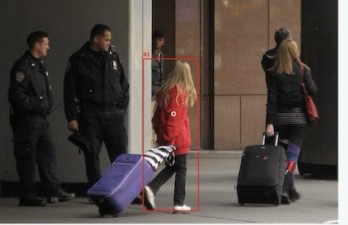

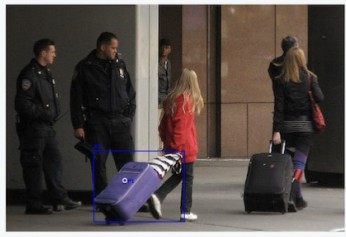

Q: Grey shirt cut off by pic on left

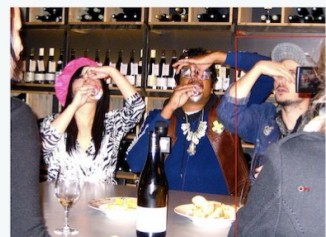

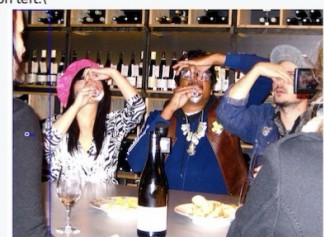

Figure 4: Segmentation qualitative results on RefCOCO datasets showing progressive refinement across TTRL steps. The model demonstrates improved boundary detection and better handling of complex object shapes, validating the effectiveness of iterative adaptation for visual reasoning tasks.

## 4.6 QUALITATIVE RESULTS

Our qualitative analysis demonstrates that VR-TTRL effectively refines both segmentation and counting predictions through iterative adaptation. Figure 4 shows progressive improvement in segmentation boundary detection and complex object handling across VR-TTRL steps on RefCOCO datasets. Figure 5 illustrates enhanced counting accuracy on CountBench, where the model achieves better outcomes through VR-TTRL refinement even with identical reasoning prompts, demonstrating meaningful improvements in task understanding and execution. More results in the Appendix B.

Q: How many pigs are there in the image?

| VisionReasoner | VisionReasonerTTRL |
|---|---|
| Thinking: " The task involves identifying the number of pigs in the image and providing their bounding boxes. I will carefully examine the image to count the pigs and identify their positions." | Thinking: " The task involves identifying the number of pigs in the image and providing their bounding boxes. I will carefully examine the image to count the pigs and identify their positions." |

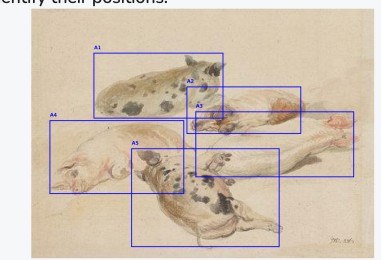

Q: How many stickers are there in the image?

| VisionReasoner | VisionReasonerTTRL |
|---|---|
| Thinking: " The task involves identifying the number of stickers in the image and comparing the objects to find the most closely matched ones. I will count the stickers and then analyze their similarities and differences." | Thinking: " The task involves identifying the number of stickers [...] The image contains multiple stickers, each with a unique design related to space aliens and extraterrestrial themes. The stickers are arranged in a grid-like pattern, and each one is distinct in its design and text." |

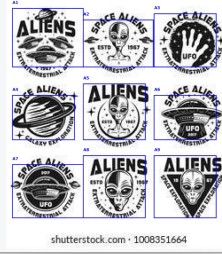

Figure 5: Counting qualitative results on CountBench illustrating enhanced accuracy through TTRL adaptation. The model achieves better counting outcomes through iterative refinement, demonstrating improved task understanding and execution even with identical reasoning prompts.

## 5 CONCLUSIONS

We introduced VR-TTRL, to our knowledge, the first framework to apply Test-Time Reinforcement Learning to vision-language models for visual reasoning tasks, enabling adaptation from a single unlabeled sample without ground truth answers. Our approach extends TTRL from language-only mathematical reasoning to visual reasoning through majority voting across model rollouts, demonstrating that effective adaptation can be achieved using only one unlabeled test sample.

Our experimental results validate the feasibility of applying TTRL to vision-language tasks through single-sample optimization without ground truth supervision. VisionReasoner-7B achieved 71.17% and 80.50% on segmentation and counting respectively, while Qwen2.5-VL-7B improved from 47.48% to 55.34% on segmentation (+7.86%) and from 41.33% to 47.50% on counting (+6.17%). The vote-only consensus baseline confirms these gains stem from RL-driven adaptation rather than simple selection, with VR-TTRL outperforming consensus-only by significant margins (e.g., +10.96% for Qwen2.5-VL-7B segmentation).

Our ablation studies revealed that adaptation effectiveness depends more on update iterations than generation diversity, with 10 updates per image providing optimal performance. The anti-consensus experiment (VR-TTRL (-)) validates that consensus quality matters—selecting the negative of consensus substantially degrades performance, confirming our gains result from principled majority voting rather than spurious optimization. While both tested models are from the Qwen family, broader cross-family validation (e.g., LLaVA-Next, Llama-3-Vision) remains valuable future work. This work demonstrates that test-time reinforcement learning provides a viable approach for adapting vision-language models to visual reasoning tasks, opening promising directions for self-supervised adaptation that enables models to better leverage their pre-trained capabilities during inference.

## 6 REPRODUCIBILITY STATEMENT

To ensure reproducibility of our results, we provide comprehensive details across multiple components of this work. Our novel VR-TTRL framework implementation, including the majority voting mechanism for vision-language model rollouts and a single-sample optimization approach, will be made available publicly. The complete experimental setup, including hyperparameter configurations, model architectures, and training procedures, will also be provided to ensure reproducibility of our findings. All datasets used in our experiments (RefCOCO, RefCOCO+, RefCOCOg, ReasonSeg, PixMo-Count, and CountBench) are publicly available. The ablation studies presented in Tables 5 and 4 include all hyperparameter settings that differ from the original setting for replication.

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

# A    EVALUATION METRICS

This appendix provides detailed definitions of the evaluation metrics used throughout our experiments.

**gIoU (Generalized Intersection over Union)**: For segmentation tasks, gIoU measures the overlap between predicted and ground truth bounding boxes or segmentation masks. It is computed as the intersection area divided by the union area of the predicted and ground truth regions, providing a scale-invariant measure of spatial accuracy. gIoU values range from 0 to 1, where 1 indicates perfect overlap and 0 indicates no overlap.

**cIoU (Complete Intersection over Union)**: cIoU extends the standard IoU metric by incorporating additional geometric information such as center point distance and aspect ratio. This metric provides a more comprehensive evaluation of bounding box quality by penalizing predictions that have good overlap but poor geometric alignment. cIoU is particularly useful for evaluating the precision of object localization in visual reasoning tasks. bbox_AP (Bounding Box Average Precision): bbox_AP measures the average precision of bounding box predictions across different IoU thresholds, typically ranging from 0.5 to 0.95 in increments of 0.05. This metric evaluates both the accuracy of object detection and the quality of bounding box localization, providing a comprehensive assessment of the model's ability to identify and localize objects in images.

**Counting Accuracy**: For counting tasks, counting accuracy is defined as the percentage of images where the number of bounding boxes extracted from the model's output exactly matches the ground truth count. This metric requires the model to both correctly identify and localize objects (by generating appropriate bounding boxes) and accurately enumerate them. We extract the count by measuring the length of the bounding box list in the model's response, which ensures that the model must not only reason about the number of objects but also successfully identify and localize each object to generate the corresponding bounding boxes. Counting accuracy is computed as the ratio of correctly counted images to the total number of test images, expressed as a percentage.

Table 6: Failure analysis for VisionReasoner-7B w/ VR-TTRL and Qwen2.5-VL-7B w/ VR-TTRL . We report the percentage of samples for which performance (IoU/accuracy) decreases after applying VR-TTRL , relative to the base model. Each threshold (0%, 1%, 5%, 10%) specifies the minimum magnitude of degradation that counts as a regression—for example, the 10% threshold measures the proportion of samples where performance drops by more than 10%. Higher thresholds therefore ignore small fluctuations and capture only substantial regressions. Overall, when VR-TTRL does not improve performance, the declines are generally minor.

| Model | Threshold | Segmentation | | | | Avg. | Counting | | | Avg. |
|---|---|---|---|---|---|---|---|---|---|---|
| | | ReasonSeg | RCO | RCO+ | RCOg | | Pixmo | | Count | |
| | | val   test | testA | testA | test | | val   test | | test | |
| Qwen2.5-VL-7B w/ VR-TTRL | 0% | 28.50  26.50 | 18.50 | 28.50 | 29.00 | 26.20 | 4.00  5.00 | | 1.50 | 3.5 |
| | 1% | 12.00  11.50 | 6.00 | 8.50 | 13.00 | 10.20 | 4.00  5.00 | | 1.50 | 3.5 |
| | 5% | 9.50  9.50 | 5.50 | 8.50 | 12.00 | 9.00 | 4.00  5.00 | | 1.50 | 3.5 |
| | 10% | 9.00  8.00 | 5.50 | 7.50 | 12.00 | 8.40 | 4.00  5.00 | | 1.50 | 3.5 |
| VisionReasoner-7B w/ VR-TTRL | 0% | 42.00  38.50 | 39.50 | 47.00 | 41.00 | 41.6 | 3.00  3.00 | | 1.00 | 2.33 |
| | 1% | 16.00  9.50 | 5.50 | 6.00 | 8.50 | 9.10 | 3.00  3.00 | | 1.00 | 2.33 |
| | 5% | 10.00  6.50 | 4.00 | 3.50 | 5.50 | 5.9 | 3.00  3.00 | | 1.00 | 2.33 |
| | 10% | 9.00  5.00 | 2.50 | 2.50 | 3.50 | 4.5 | 3.00  3.00 | | 1.00 | 2.33 |

## B ADDITIONAL RESULTS

Analysis in Table 6 demonstrates that when our method fails to improve results, the performance oscillates around the baseline prediction rather than causing significant degradation.

Table 7: Rollout trade-off results on RefCOCOg using VisionReasoner (unscaled values). Increasing rollout size from 8 to 32 provides minimal performance gains (73.19 to 73.23 gIoU) but triples computational cost (86.7 to 229.8 seconds per image), indicating that small rollouts are sufficient for effective consensus building.

| Rollout | gIoU | cIoU | bbox_AP | Time (s/img)↓ |
|---|---|---|---|---|
| Baseline | 71.66 | 68.39 | 88.50 | - |
| 8 | 73.19 | 69.16 | 89.00 | 86.7 (±4.8) |
| 16 | 73.10 | 68.65 | 89.50 | 132.4 (±4.7) |
| 32 | 73.23 | 69.20 | 89.00 | 229.8 (±5.4) |

Table 8: Effect of #updates per image on ReasonSeg test using VisionReasoner (unscaled values, rollout size 8). Performance peaks at 10 updates (62.94 gIoU, 54.99 cIoU) before diminishing returns, demonstrating optimal adaptation without overfitting to the single sample.

| #Updates | gIoU | cIoU | Time (s/img) |
|---|---|---|---|
| 0 | 59.65 | 50.60 | 0 |
| 1 | 59.65 | 50.60 | 20.81 |
| 2 | 59.84 | 50.27 | 41.86 |
| 3 | 61.95 | 53.87 | 62.73 |
| 5 | 62.59 | 54.39 | 104.53 |
| 10 | 62.94 | 54.99 | 206.42 |
| 15 | 62.44 | 54.24 | 306.86 |

We evaluate three threshold configurations: loose (IoU=0.3, bbox L1=20px, point L1=50px), base (IoU=0.5, bbox L1=10px, point L1=30px) and strict (IoU=0.7, bbox L1=5px, point L1=15px).

## C ADDITIONAL TEST-TIME ADAPTATION BASELINES

Table 9: Effect of reward threshold settings on gIoU and cIoU using VisionReasoner w/ VR-TTRL on RefCOCOg. Three configurations are compared: loose (IoU=0.3, bbox L1=20px, point L1=50px), base (IoU=0.5, bbox L1=10px, point L1=30px), and strict (IoU=0.7, bbox L1=5px, point L1=15px). Baseline VisionReasoner results for reference.

| Configuration | gIoU | cIoU |
|---|---|---|
| Baseline | 71.66 | 68.39 |
| Loose (IoU=0.3) | 72.43 | 67.59 |
| Base (IoU=0.5) | 73.19 | 69.16 |
| Strict (IoU=0.7) | 72.72 | 68.08 |

## C.1 IMPLEMENTATION DETAILS

We consider four single-sample test-time adaptation baselines, all operating under the same setting as VR-TTRL : each test image–query pair is adapted independently, without access to other test samples, and we update the same parameter subset as in VR-TTRL (language-side and adapter parameters, with the vision encoder frozen).

**Tent-style entropy minimization.** For each test image, we generate an answer with the current model and minimize the token-level entropy of the answer distribution. Concretely, let $p_\theta(y_t \mid x)$ denote the distribution over answer tokens at position $t$. We perform $K$ gradient steps on the loss

$$\mathcal{L}_{\text{tent}} = \sum_{t \in \mathcal{T}_{\text{ans}}} H(p_\theta(y_t \mid x)),$$

where $\mathcal{T}_{\text{ans}}$ indexes the tokens corresponding to the structured answer (e.g., JSON fields). The vision encoder is frozen.

**MEMO-style consistency TTA.** Given a test image, we construct $K$ stochastic augmentations (random crop/flip/color jitter) and obtain answer distributions $\{p_\theta^{(k)}\}_{k=1}^K$. We then minimize the inconsistency to the mean prediction,

$$\mathcal{L}_{\text{memo}} = \frac{1}{K} \sum_{k=1}^K \text{KL}\left(p_\theta^{(k)} \parallel \bar{p}_\theta\right), \quad \bar{p}_\theta = \frac{1}{K} \sum_{k=1}^K p_\theta^{(k)},$$

again only over answer tokens.

**SFT (single greedy pseudo-label).** The model first produces a single greedy answer for the test sample, which is treated as a hard pseudo-label $\hat{y}$. We then perform $K$ steps of supervised fine-tuning using the standard negative log-likelihood loss

$$\mathcal{L}_{\text{sft}} = -\sum_t \log p_\theta(\hat{y}_t \mid x, \hat{y}_{<t}).$$

**SFT (majority pseudo-label).** We sample multiple rollouts for the test sample and construct a pseudo-label by majority voting on the structured output (segmentation / count), using the same consensus mechanism as in VR-TTRL . This pseudo-label is then used in the same supervised objective as above. Unlike VR-TTRL , this baseline does not use RL or relative rewards; it is a non-RL counterpart that still exploits majority-vote pseudo-supervision.

## C.2 QUANTITATIVE RESULTS

Table 10: Single-sample test-time adaptation baselines on ReasonSeg and RefCOCOg.

| Method | ReasonSeg val | RefCOCOg testA |
|---|---|---|
| **Qwen2.5-VL-7B** | | |
| Base (no adaptation) | 41.89 | 50.46 |
| + Tent (entropy TTA) | 38.71 | 39.43 |
| + MEMO (consistency TTA) | 40.12 | 41.29 |
| + SFT (single greedy pseudo-label) | 35.73 | 36.43 |
| + SFT (majority pseudo-label) | 43.17 | 46.73 |
| **+ VR-TTRL** | **50.55** | **56.38** |
| **VisionReasoner-7B** | | |
| Base (no adaptation) | 66.08 | 71.66 |
| + Tent (entropy TTA) | 58.73 | 62.77 |
| + MEMO (consistency TTA) | 63.93 | 65.12 |
| + SFT (single greedy pseudo-label) | 62.71 | 64.25 |
| + SFT (majority pseudo-label) | 64.67 | 68.13 |
| **+ VR-TTRL** | **66.28** | **73.19** |

Table 11: Offline vs. test-time adaptation with the same consensus+format GRPO objective on Qwen2.5-VL-7B.

| Regime | Adaptation data | ReasonSeg val (gIoU) | RefCOCOg testA (cIoU) |
|---|---|---|---|
| Base (no adaptation) | – | 41.89 | 50.46 |
| Offline (train-only) | train split | 47.65 | 53.27 |
| VR-TTRL (test-time, ours) | test ($1\times$ per image) | **50.55** | **56.38** |

Table 10 summarizes the performance of these baselines on ReasonSeg and RefCOCOg for both backbones. All methods adapt each test image independently under the same computational budget as VR-TTRL .

## D   OFFLINE VS. TEST-TIME ADAPTATION

To disentangle the effect of the VR-TTRL  objective from the effect of adapting on the test distribution, we perform an ablation that keeps the *objective* and the *trainable parameters* fixed and only changes which data distribution is used for adaptation.

Concretely, we compare three regimes for **Qwen2.5-VL-7B** on ReasonSeg and RefCOCOg: (i) the base model without any adaptation, (ii) an *offline* variant where we optimize the same consensus+format GRPO objective on the *training split*, and (iii) the proposed VR-TTRL  method, which applies the same objective at test time and adapts the model separately for each test image–query pair.

Offline training with the same objective already improves over the base model (e.g., from 41.89 to 47.65 gIoU on ReasonSeg), confirming that the consensus+format GRPO objective itself is beneficial. However, VR-TTRL  still achieves the best performance (50.55 gIoU and 56.38 cIoU), showing that adapting on the *test* distribution, per sample, provides additional gains beyond offline training on the original data. In other words, when the objective and trainable parameters are held fixed, changing the adaptation distribution from train to test yields further improvements, highlighting the specific role of test-time adaptation rather than consensus selection alone.

Q: A purple suitcase, being pulled by a woman with blonde hair

### VisionReasoner

Thinking: " The task involves identifying a purple suitcase being pulled by a woman with blonde hair. I need to locate the woman with blonde hair and the purple suitcase in the image. The woman with blonde hair is wearing a red jacket and is positioned in the foreground, pulling a purple suitcase. The purple suitcase is clearly visible and matches the description."

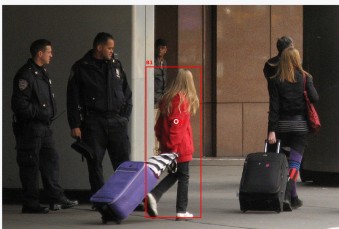

### VisionReasonerTTRL

Thinking: " The task involves identifying a specific object: a purple suitcase being pulled by a woman with blonde hair. I need to locate this object in the image and provide its bounding box coordinates."

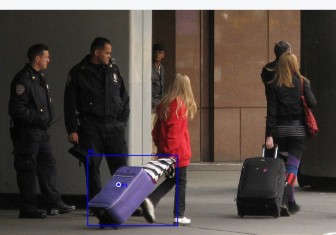

Q: Grey shirt cut off by pic on left

### VisionReasoner

Thinking: " The task involves identifying a person wearing a grey shirt who is partially cut off by the picture on the left side. I need to locate the person in the image who matches this description and provide their bounding box coordinates."

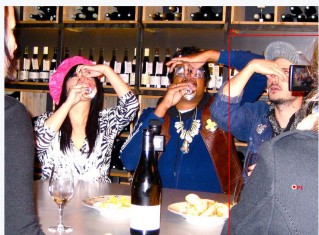

### VisionReasonerTTRL

Thinking: " The task involves identifying a specific person in the image based on their clothing and position, and then finding the most closely matched object. [...] The person on the far left of the image appears to be wearing a dark grey or black shirt, which matches the description of the grey shirt mentioned in the instruction. The person is also partially cut off by the edge of the image, which aligns with the description of being \"cut off by pic on left.\""

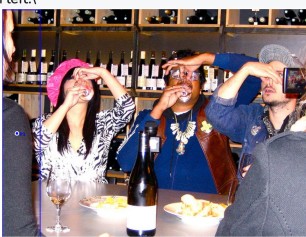

Q: Mans arm hanging

### VisionReasoner

Thinking: " The task involves identifying the person whose arm is hanging and then finding the most closely matched object. The image contains two men sitting at a table, and one of them has his arm hanging off the table. The man on the right has his arm hanging, while the man on the left does not. The man on the right is wearing a blue shirt and is positioned on the right side of the image."

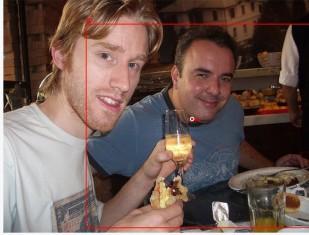

### VisionReasonerTTRL

Thinking: " The task involves identifying the person whose arm is hanging and comparing it to the other individuals in the image. The person on the right side of the image appears to have their arm hanging off the table, which is a distinguishing feature. The other two individuals are seated and their arms are not visible or hanging."

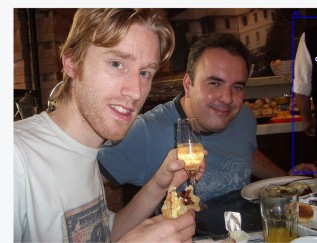

Q: A giraffe that is traveling to the left of the frame and looking to the right of the frame

### VisionReasoner

Thinking: " The task involves identifying a giraffe that is moving to the left of the frame and looking to the right of the frame. I need to analyze the direction of the giraffe's head and its position relative to the frame. The giraffe on the left appears to be looking to the right, and it is positioned to the left of the frame. The other two giraffes are not aligned with the criteria as they are either not moving or not positioned to the left of the frame."

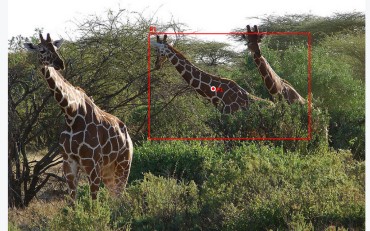

### VisionReasonerTTRL

Thinking: " I need to identify the giraffe that is traveling to the left of the frame and looking to the right of the frame. The giraffe in the foreground on the left side of the image is facing right and appears to be moving left, which matches the description. The other two giraffes are either facing forward or to the right, so they do not match the description."

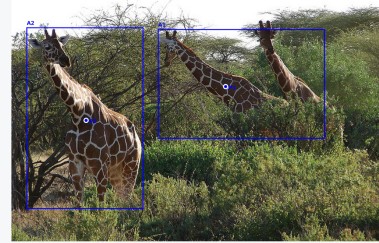

Figure 6: Segmentation qualitative results on RefCOCO datasets showing progressive refinement across TTRL steps. The model demonstrates improved boundary detection and better handling of complex object shapes, validating the effectiveness of iterative adaptation for visual reasoning tasks.

