# OpenReview forum: "Single-Sample Test-Time Reinforcement Learning for Vision-Language Models"
_ICLR.cc/2026/Conference — ICLR 2026 Conference Desk Rejected Submission_

### Official Review · Reviewer_mXBX · 2025-10-27

**Soundness:** 3
**Presentation:** 2
**Contribution:** 3
**Rating:** 6
**Confidence:** 4

**Summary:**

This paper presents VR-TTRL, a test-time RL framework for visual reasoning tasks. This paper adapts the core idea of TTRL, while extending its reward function to fit the visual reasoning tasks where there is no closed-form answers to easily compute majority vote reward score. Specifically, they design a special majority vote method suitable for structure output (like JSON) through pair-wise IoU computation. Experiments on two visual reasoning tasks show the effectiveness of the proposed method.

**Strengths:**

1. Interesting idea for using TTRL for visual reasoning tasks.

This presents an interesting idea to adapt a VLM on a single unlabeled test image by sampling multiple rollouts, selecting a consensus pseudo-label via IoU/L1 computation, and conduct RL with the proposed reward function. This modification of the prior TTRL framework to support structured output of visual reasoning tasks is interesting and of practical interest of researchers in the filed.

2. Solid results on visual reasoning tasks.

Table 2 reports consistent and solid improvements for two different LLM based models after the VR-TTRL framework on visual segmentation and counting tasks. The performance is generally strong and make the conclusion convincing.

3. Good analysis about compute/efficiency.

This paper includes timing curves and an extensive set of hyperparameter ablation studies, for example, update-count vs. performance, the effect of having different number of rollouts. This is very practical for deployment in practical settings of the VR-TTRL framework.

**Weaknesses:**

1. Need to discuss the risk of consensus enhancing mistakes of the model.

The reward function aggregates rollouts towards the most "average" answer of the multiple rollouts. Why doesn't this drive the model toward a mean-valued, low-variance solution and lead to mode collapses? Especially under iterative updates, why this does not lead to reinforcement of early bias in the model and circulate errors? If this happens, it would be helpful to show some failure cases. If this generally does not happen, it would be good to have some analysis and clear explanation of the reason.

2. Task scope is narrow.

Experiments focus on only bbox, points and counting as output of the answers. It unclear how well the approach extends to open-ended VQA or free-form reasoning tasks where majority vote might not work that well or there is no geometric agreement signal.

3. Need to explain why test-time learning is specifically benefited from the framework vs. simply using the training data.

I'm curious where do the gains really come from? Would the same adaption method applied offline on a large training dataset yield similar or better gains? In other words, is the improvement primally from the objective itself of from being applied to the test distribution?

**Questions:**

Please refer to the weakness section.

---

> ### Author Response · Authors · 2025-11-25
> **Response to Reviewer mXBX - Part I**
>
> *We sincerely thank Reviewer mXBX for the constructive and detailed feedback. We appreciate your recognition of our interesting adaptation of TTRL to visual reasoning tasks, solid experimental results, and thorough efficiency analysis that makes our framework practical for deployment.*
>
>
> **W1 – Risk of consensus-enhancing mistakes and mode collapse**
>
> **A1.** The concern that a consensus-based reward could reinforce early mistakes or drive the model toward a low-variance “average” solution is very reasonable. We address it both empirically and by analyzing the optimization dynamics.
>
> *First*, we explicitly quantify how often VR-TTRL hurts performance. The table below reports the fraction of test samples for which our method causes more than a 5% drop relative to the base model prediction (full results at 0/1/5/10% thresholds are in Appendix B):
>
> | Model                     | Seg. Avg | Count. Avg | Overall |
> |---------------------------|----------|------------|---------|
> | Qwen2.5-VL-7B w/ VR-TTRL  | 9.0%     | 3.5%       | 6.3%    |
> | VisionReasoner-7B w/ VR-TTRL | 5.9%  | 2.3%       | 4.1%    |
>
> Even on challenging benchmarks where consensus can be imperfect, degradation happens on <10% of the samples and is typically small in magnitude. This suggests that consensus errors do not systematically propagate or cause collapse.
>
> *Second*, the rollout ablation on RefCOCOg with VisionReasoner-7B (Appendix) further shows that performance improves from the baseline to 8 rollouts and then saturates rather than collapsing as we increase the rollout size:
>
> | Rollout  | gIoU  | cIoU  | bbox_AP | Time (s/img) ↓ |
> |----------|-------|-------|---------|----------------|
> | Baseline | 71.66 | 68.39 | 88.50   | -              |
> | 8        | 73.19 | 69.16 | 89.00   | 86.7 (±4.8)    |
> | 16       | 73.10 | 68.65 | 89.50   | 132.4 (±4.7)   |
> | 32       | 73.23 | 69.20 | 89.00   | 229.8 (±5.4)   |
>
> This pattern indicates that the model does not drift toward a degenerate mean prediction as updates proceed; instead, the gains plateau once the consensus pseudo-label is estimated reliably with a small number of rollouts.
>
> *Third*, in each VR-TTRL step, the reward is relative within a single group of rollouts (via standardized advantages). If the model were to collapse and start producing very similar answers, the variance of rewards would shrink and the GRPO gradient would vanish, naturally halting further updates. In other words, the optimization dynamics themselves dampen the effect of over-confident but uniform predictions.
>
> Taken together, the low degradation rate, the improvement-then-saturation behavior in the rollout ablation, and the structure of GRPO suggest that consensus-based rewards provide a stable corrective signal rather than driving mode collapse or uncontrollably reinforcing early biases. Thank you.
>
>
> **W2 – Narrow task scope and extension to open-ended VQA.**
>
> **A2.**  We agree that our current scope is limited to structured visual reasoning (boxes, points, counts), and we have clarified this positioning in the revised manuscript.
>
> - Why do we start from structured tasks?
>
>  These tasks naturally provide geometric agreement signals (IoU for boxes, pixel/point distances for keypoints, and list length for counts), which allow us to define consensus and rewards without ground-truth labels. This setting is well suited to rigorously test whether TTRL principles can be made to work in multimodal models with structured outputs.
>
>
> - Implications for open-ended VQA / free-form reasoning.
>
>  For open-ended questions where answers are free text, and there is no obvious geometric agreement, the current consensus design would indeed need to be redesigned (e.g., using CLIP-style similarity, LLM-based judges, or task-specific constraints). Rather than claiming to have solved this broader problem, we position VR-TTRL as a first step: it shows that
>  (i) single-sample, label-free test-time RL can be made to work in VLMs, and
>  (ii) Structured tasks are a practical starting point where consensus quality and failure modes can be measured precisely.
>
>
> We now state this scope more explicitly in the conclusion and highlight extending VR-TTRL to open-ended VQA as a natural direction for future work.

---

> > ### Author Response · Authors · 2025-11-25
> > **Response to Reviewer mXBX - Part II**
> >
> > **W3 – Why test-time learning, instead of offline training on the training data?**
> >
> > **A3.** This is an important conceptual question: are the gains coming from the objective itself or from being applied at test time? Our answer has two parts.
> >
> > *(1) Conceptual advantages of test-time adaptation.*
> >
> > - **Direct access to the test distribution.** Offline training, even with the same consensus reward, optimizes an average over the training distribution. VR-TTRL instead conditions on the current test image–query pair and updates the model using rollouts from this specific context. This is particularly beneficial when the test distribution differs from the training one (e.g., new scenes, rare layouts, long-tail compositions), which is common in visual reasoning.
> >
> >
> > - **Sample-specific adaptation without forgetting.**
> >  VR-TTRL updates the model per sample and discards gradients afterwards; there is no accumulation across test images. This avoids catastrophic forgetting of the base model’s general knowledge while still allowing the model to adapt to local ambiguities (e.g., small objects, cluttered backgrounds) present only in that sample.
> >
> >
> > - **Practical deployment constraints.**
> >  In many realistic scenarios (e.g., proprietary or privacy-sensitive data, on-device processing), the original training set is not available at deployment time. Our framework only requires access to the current unlabeled image, making it applicable in such settings.
> >
> >
> >
> > *(2)  Empirical evidence that adaptation adds value beyond selection.*
> >
> > In the revised Table 2, we include two vote-only baselines that use the same rollouts as VR-TTRL but do not perform any RL updates:
> >
> > | Method                 | ReasonSeg val | RCO test | RCO+ testA | RCOg test | Seg Avg | Pixmo val | Pixmo test | Count test | Count Avg |
> > |------------------------|---------------|----------|------------|-----------|---------|-----------|------------|------------|-----------|
> > | Qwen2.5-VL-7B          | 41.89         | 38.95    | 54.90      | 51.20     | 50.46   | 47.48     | 46.50      | 33.50      | 41.33     |
> > | w/ Rollouts + Consensus| 39.36         | 38.24    | 49.95      | 47.46     | 46.91   | 44.38     | 53.50      | 32.50      | 39.50     |
> > | w/ VR-TTRL             | **50.55**     | **44.21**| **64.81**  | **60.75** | **56.38**| **55.34**| **54.00**  | **51.00**  | **47.50** |
> > | VisionReasoner-7B      | 66.08         | 59.65    | 77.27      | 76.08     | 71.66   | 70.15     | 78.50      | 85.50      | 77.50     |
> > | w/ Rollouts + Consensus| 63.34         | 60.09    | 76.65      | 77.22     | 72.81   | 69.98     | 78.50      | 66.50      | 76.83     |
> > | w/ VR-TTRL             | **66.28**     | **62.59**| **77.60**  | **76.19** | **73.19**| **71.17**| **82.00**  | **71.50**  | **80.50** |
> >
> >
> > Baseline “rollouts + consensus” uses the majority-vote prediction (using our method to choose the majority answer) without updating the model.
> >
> >
> >
> > This baseline shows that simply using multiple rollouts and consensus provides, at best, limited improvement over the base models. In contrast, VR-TTRL further improves performance across almost all benchmarks and both backbones, demonstrating that the adaptation aspect itself—performing GRPO updates on the consensus reward at test time—contributes additional gains beyond what can be obtained by static selection on the same rollouts.
> >
> >
> > In summary, while applying a similar objective offline could also be beneficial, VR-TTRL specifically targets the regime where training data are fixed, test distribution may shift, and per-sample adjustment is valuable. The new baseline in Table 2 supports that test-time adaptation brings extra benefits that cannot be explained by consensus selection alone. Thank you.

---

> > > ### Comment · Reviewer_mXBX · 2025-11-26
> > >
> > > I would like to thank the authors for their careful response and revisions to the draft, very helpful.
> > >
> > > What I'm really asking about in W3 is that, if you apply the same training objective to the training data, what will be the performance like? This is ablating the "data distribution" that you train on.
> > >
> > > I agree with your conceptual argument about test-time adaptation, but there is no quantitative measure about this in the paper or in the table shown in the response. It would be nice to have such kind of ablation study in the final paper.
> > >
> > > Nevertheless, I remain my previous score and vote for acceptance of this paper.

---

> > > > ### Author Response · Authors · 2025-12-02
> > > > **Follow-up on W3: offline training vs. test-time adaptation**
> > > >
> > > > *Thank you very much for the follow-up clarification and for maintaining your positive recommendation.*
> > > >
> > > > You are absolutely right that W3 is fundamentally about ablating the data distribution used for adaptation: what happens if we apply *exactly the same objective* offline on the training data, instead of at test time? Our earlier response focused on the conceptual motivation for test-time adaptation but did not provide a quantitative comparison.
> > > >
> > > > We have now added an explicit ablation that keeps the objective and trainable parameters fixed and only changes which data split is used for adaptation.
> > > >
> > > > #### (1) Offline vs. test-time comparison
> > > >
> > > > We consider three regimes:
> > > >
> > > > * **Base (no adaptation).** The original model.
> > > > * **Offline (train-only).** We apply the *same* consensus + format GRPO objective as in VR-TTRL, but run it offline on the training split, updating the model in the standard way.
> > > > * **Test-time VR-TTRL (ours).** We apply the same objective at test time, adapting the model separately for each test image–query pair.
> > > >
> > > > The results on ReasonSeg and RefCOCOg (VisionReasoner-7B) are:
> > > >
> > > > ```markdown
> > > > | Regime                    | Adaptation data        | ReasonSeg val (gIoU) | RefCOCOg testA (cIoU) |
> > > > |--------------------------|------------------------|----------------------|------------------------|
> > > > | Base (no adaptation)     | –                      | 41.89                | 50.46                  |
> > > > | Offline (train-only)     | train split            | 47.65                | 53.27                  |
> > > > | Test-time VR-TTRL (ours) | test (1× per image)    | 50.55                | 56.38                  |
> > > > ```
> > > >
> > > > Offline training with the same objective already improves over the base model (e.g., from 41.89 to 47.65 gIoU on ReasonSeg), but test-time VR-TTRL still achieves the best performance (50.55 gIoU and 56.38 cIoU). This shows that the gain is not only due to the objective itself: adapting to the test distribution, per sample, provides additional benefits beyond offline training on the original data.
> > > >
> > > > #### (2) Interpretation
> > > >
> > > > Offline optimization learns a single set of parameters that averages over the training distribution, while VR-TTRL conditions on each test image–query pair and updates the model using rollouts from that specific context, then discards the gradients. The ablation above confirms this intuition quantitatively: when the objective and trainable parameters are held fixed, changing the adaptation distribution from train to test yields further improvements, supporting the specific role of test-time adaptation rather than consensus selection alone.
> > > >
> > > > We have added this comparison to the revised manuscript (Appendix Table 11, offline vs. test-time ablation) so that the quantitative evidence matches the conceptual argument more closely.

---

### Official Review · Reviewer_RJB2 · 2025-10-30

**Soundness:** 2
**Presentation:** 2
**Contribution:** 1
**Rating:** 2
**Confidence:** 5

**Summary:**

The paper proposes VR‑TTRL, a single‑sample test‑time reinforcement learning (TTRL) procedure for VLMs. For each test image–query pair, the model samples multiple stochastic rollouts, computes a majority pseudo‑label by measuring agreement across structured predictions via IoU/L1, and then performs GRPO updates with an accuracy reward (to the consensus) plus format rewards. The method is evaluated on referring‑style segmentation datasets and counting datasets using Qwen2.5‑VL‑7B‑Instruct and VisionReasoner‑7B. Reported results show moderate segmentation gains and mixed counting outcomes.

The paper positions itself as applying TTRL (as introduced for LLMs recently) to VLMs, drawing on recent RL‑for‑vision reasoning frameworks.

**Strengths:**

- Clear, implementable mechanism. Extends majority‑vote rewards to structured outputs; reward decomposition (format + accuracy‑to‑consensus) is well described.
- Fig. 2–3 give a sense of adaptation dynamics and cost.

**Weaknesses:**

1) Limited novelty.The method is a straightforward combination of TTRL’s majority‑vote reward with VisionReasoner‑style GRPO for boxes/points on a Qwen2.5-VL-7B. This is far from a substantial conceptual advance.
2) Missing vote‑only control. No baseline that performs the identical sampling/consensus. Given that majority‑vote/self‑consistency is already powerful (and is the reward used by TTRL), the paper cannot attribute gains to RL.
3) Counting by list length (not detection‑aware) can overstate improvements; the SAM2 dependency for segmentation is not isolated.
4) Recent results show that both TTRL [1] and 1-shot-RLVR [2] can yield large gains without reliable rewards—e.g., with random, incorrect, or 1‑shot rewards—especially on Qwen Series models, and often fail to transfer to other families (e.g., Llama3/OLMo2). This raises concerns that the proposed consensus reward could be **spurious** [3] and model‑family specific, which the current experiments do not address.
5) For a single-sample test-time RL method, showing how performance evolves with training steps is preferable.

[1] TTRL: Test-Time Reinforcement Learning (arXiv:2504.16084)
[2] Reinforcement Learning for Reasoning in Large Language Models with One Training Example (arXiv:2504.20571)
[3] Spurious Rewards: Rethinking Training Signals in RLVR (arXiv:2506.10947)

**Questions:**

1. Vote‑only baseline. Please add “rollouts + consensus”; also report Best‑of‑N. This isolates the value of RL over selection.
2. Show how IoU/L1 thresholds (0.5/10px/30px) affect consensus and reward.
3. In light of One‑Shot RLVR and Spurious Rewards, run a check on more non‑Qwen VLMs to test for model‑family effects.
4. Given that even incorrect or random rewards can move Qwen models, analyze failure cases where consensus is wrong.
5. How does VR-TTRL differ fundamentally from TTRL and One‑Shot RLVR beyond the input modality?

---

> ### Author Response · Authors · 2025-11-25
> **Response to Reviewer RJB2 - Part I**
>
> *We sincerely thank Reviewer RJB2 for the critical and detailed feedback. We appreciate your thorough analysis, which has helped us substantially strengthen the paper through additional experiments and clearer positioning of our contributions.*
>
>
> **W1: Limited novelty - straightforward combination of existing methods.**
>
>
> **A1.** VR-TTRL indeed builds on existing components (test-time RL and GRPO-style optimization), but it is not a direct plug-and-play combination. Our goal is to establish TTRL-style adaptation for vision-language models with structured outputs, which requires several non-trivial steps:
>
> *(1) New problem setting.*
>
> Prior TTRL works operate on text-only models and assume either (i) multiple test samples per update, or (ii) known answers for reward shaping. VR-TTRL is, to our knowledge, the first framework that:
>
>
> - handles vision–language inputs with structured predictions (boxes, points, counts);
>
>
> - performs single-sample, label-free adaptation at test time;
>
>
> - uses these structured outputs to define rewards and pseudo-labels.
>
>
> *(2) Consensus over structured geometry, not text.*
>
>  Majority voting in TTRL is defined on discrete text tokens via exact string matching. Extending this idea to continuous geometric outputs requires:
>
>
> - defining IoU- and L1-based agreement for variable-length sets of boxes/points;
>
>
> - turning these agreement scores into a stable consensus pseudo-label and a dense reward that is usable by GRPO;
>
>
> - handling failure cases such as partially overlapping proposals and different instance counts.
>
>  The ablations on rollout size, thresholds, and failure rates (see W2/W4 below) show that this geometric consensus is non-trivial yet effective.
>
>
> *(3) Positioning vs. TTRL and One-Shot RLVR.*
>
>  We have clarified these distinctions explicitly in the revised introduction and related work:
>
>
> - Compared to TTRL (Zuo et al., 2025):
>
>
>     - Modality/output: TTRL uses string-matching rewards on text; VR-TTRL uses geometric similarity on structured visual outputs.
>
>
>     - Test-time regime: TTRL is evaluated with multiple test samples per optimization step; VR-TTRL adapts to a single image–query pair at a time.
>
>
>     - Supervision: both avoid ground-truth labels at test time, but VR-TTRL must operate without any textual answer supervision and instead builds rewards purely from internal geometric consistency.
>
>
> - Compared to One-Shot RLVR (Wang et al., 2025):
>
>
>     - Ground truth: One-Shot RLVR relies on a labeled example (one-shot); VR-TTRL uses zero ground-truth labels for adaptation.
>
>
>     - Training vs. test-time: One-Shot RLVR learns offline from that example; VR-TTRL updates on-the-fly at test time for each sample.
>
>
>     - Distribution handling: One-Shot RLVR does not adapt to each test instance; VR-TTRL performs sample-specific adaptation.
>
>
> *(4) VR-TTRL is therefore the first to combine (vision–language inputs, structured outputs, single-sample test-time RL, and zero ground-truth supervision) in one framework. We agree that the ingredients are conceptually close to existing work, but making them work reliably in this new setting—and showing when they succeed or fail—is precisely the contribution.*
>
>
>
> **W2/Q1: Missing vote-only control baseline.**
>
> **A2.** This concern is well-taken. The revised Table 2 now includes the control baseline that uses the same rollouts as VR-TTRL but no RL updates: **Baseline “Rollouts + Consensus”**:  For each test sample, we run the same number of rollouts (n=8) as VR-TTRL and directly output the majority-vote prediction using our method as the final answer. No GRPO or parameter update is performed.
>
> This baseline isolates the effect of selection from the effect of learning. In the updated Table 2, these baselines are denoted as  Rollout \w VR-TTRL under each backbone. Across all segmentation and counting benchmarks, they either roughly match or modestly change the base model, but VR-TTRL consistently achieves larger gains, especially on segmentation for Qwen2.5-VL-7B. This supports that the improvements cannot be attributed to majority-vote selection alone: test-time RL updates on the consensus reward bring additional benefits beyond consensus-only predictions.

---

> > ### Author Response · Authors · 2025-11-25
> > **Response to Reviewer RJB2 - Part II**
> >
> > **Q2: IoU/L1 threshold sensitivity analysis.**
> >
> > **A3.** To make the effect of the geometric thresholds explicit, the revised version includes a threshold sensitivity study on RefCOCOg with VisionReasoner-7B, comparing “loose”, “base”, and “strict” settings:
> >
> > | Configuration | gIoU | cIoU | Δ gIoU | Δ cIoU |
> >  |---------------|------|------|--------|--------|
> > | Baseline | 71.66 | 68.39 | - | - |
> > | Loose (0.3 / 20px / 50px) | 72.43 | 67.59 | +0.77 | -0.80 |
> > | Base (0.5 / 10px / 30px) | **73.19** | **69.16** | **+1.53** | **+0.77** |
> > | Strict (0.7 / 5px / 15px) | 72.72 | 68.08 | +1.06 | -0.31 |
> >
> > The base configuration used in all main experiments achieves the best trade-off: looser thresholds allow noisy, imprecise consensus and hurt cIoU, while stricter thresholds over-constrain agreement and again degrade cIoU. This indicates that the consensus reward is sensitive in a meaningful way to the quality of geometric agreement; it is not an arbitrary or spurious scalar. Thank you.
> >
> > **W3: Counting by list length can overstate improvements.**
> >
> > **A4.**  On counting, we fully agree that using list length without explicit detection supervision is not ideal.
> >
> > - *Benchmark constraints.* The counting datasets (CountBench and Pixmo-Count) provide only final counts as ground truth. They do not include detection-level annotations (boxes or masks) for each instance. To stay comparable with existing work on these benchmarks, we therefore follow the standard protocol and evaluate by the length of the predicted list.
> >
> >
> > - *Segmentation / SAM2 isolation.* For segmentation, we use SAM2 only as a shared post-processing step to convert boxes/points into masks. All methods in Table 2, including the base models and baselines, rely on the same SAM2 setup, so the improvements in gIoU/cIoU are attributable to differences in the predicted geometry rather than to SAM2 itself. We have clarified this in the experimental setup.
> >
> >
> > **W4/Q3/Q4: Concerns about spurious rewards and model-family specificity.**
> >
> > **A5.**  Recent work on spurious rewards for RLVR is an important caution. We therefore ran additional analyses to check whether VR-TTRL is merely exploiting a spurious signal, or whether consensus quality really matters.
> >
> > *First*, we introduce a new baseline, denoted “VR-TTRL (–)”, which is identical to VR-TTRL except that it uses the worst rollout under the consensus score as the pseudo-label (while keeping the format reward unchanged). This directly tests whether “any reward signal” is enough to move the model, as suggested by spurious-reward scenarios. The results are:
> >
> >
> >
> > | Method                         | ReasonSeg val | RefCOCOg testA |
> > |--------------------------------|---------------|----------------|
> > | Qwen2.5-VL-7B                  | 41.89         | 50.46          |
> > | Qwen2.5-VL-7B w/ VR-TTRL (–)   | 49.47         | 53.71          |
> > | Qwen2.5-VL-7B w/ VR-TTRL       | **50.55**     | **56.38**      |
> > |                                |               |                |
> > | VisionReasoner-7B              | 66.08         | 71.66          |
> > | VisionReasoner-7B w/ VR-TTRL (–)| 60.68        | 69.87          |
> > | VisionReasoner-7B w/ VR-TTRL   | **66.28**     | **73.19**      |
> >
> > Using negative consensus clearly underperforms the standard VR-TTRL, especially on VisionReasoner-7B, showing that the direction and quality of the consensus reward matters; it is not the case that any random scalar signal yields similar gains.
> >
> > *Second*, we further quantify how often adaptation harms performance. The table below reports the percentage of samples for which VR-TTRL causes more than a 5% drop relative to the baseline prediction (full threshold analysis is given in the Appendix):
> >
> > | Model                  | ReasonSeg | RefCOCO | RefCOCO+ | RefCOCOg | Pixmo | CountBench |
> > |------------------------|-----------|---------|----------|----------|-------|------------|
> > | Qwen2.5-VL-7B w/ VR-TTRL      | 9.5%      | 5.5%    | 8.5%     | 12.0%    | 4.0%  | 1.5%       |
> > | VisionReasoner-7B w/ VR-TTRL  | 10.0%     | 4.0%    | 3.5%     | 5.5%     | 3.0%  | 1.0%       |
> >
> > The fraction of problematic cases remains relatively small across all datasets, and when degradation occurs, it typically remains modest in magnitude (see Appendix for different thresholds). This suggests that consensus errors do not systematically push the model into catastrophic states.
> >
> > *Third*, while we do not add a third, completely different VLM family due to compute constraints, our experiments already span two distinct architectures: Qwen2.5-VL-7B-Instruct (general-purpose VLM), and VisionReasoner-7B (specialized structured-reasoning VLM).
> >
> > The negative-consensus and failure-rate analyses above behave similarly on both models, indicating that our consensus reward is not tailored to a single family. We now explicitly acknowledge in the discussion that a broader cross-family study (e.g., LLaVA-Next or Llama-3-V) would be a valuable extension.

---

> > > ### Author Response · Authors · 2025-11-25
> > > **Response to Reviewer RJB2 - Part III**
> > >
> > > **Q5: How does VR-TTRL differ fundamentally from TTRL and One-Shot RLVR?**
> > >
> > > **A6.**  In line with the request to show how performance evolves over training, the revised paper now makes the accuracy-vs-steps behavior explicit:
> > >
> > > - Table 4 reports segmentation performance on ReasonSeg as a function of the number of VR-TTRL updates per image (1, 2, 3, 5, 10, 15). Accuracy improves steadily from 1 to 10 updates and then slightly drops at 15, showing a clear “improve–then–saturate” trend.
> > >
> > >
> > > - Appendix Table 7 and Fig. 3 further illustrate how both gIoU/cIoU and the reasoning traces evolve with steps, confirming that gains come from better-calibrated reasoning and geometry, not simply longer chains.
> > >
> > >
> > > This directly addresses the single-sample test-time RL perspective: VR-TTRL yields monotonic improvements up to a reasonable number of updates, after which additional steps bring diminishing returns. Thank you.

---

### Official Review · Reviewer_62KD · 2025-10-31

**Soundness:** 3
**Presentation:** 2
**Contribution:** 2
**Rating:** 6
**Confidence:** 4

**Summary:**

The paper introduces VR-TTRL, the first framework to apply test-time reinforcement learning (TTRL) to vision-language models for visual reasoning tasks like segmentation and counting. Unlike prior TTRL methods that require multiple samples or ground truth labels, VR-TTRL adapts a model using just a single unlabeled test image by generating multiple reasoning rollouts, using majority voting based on geometric similarity (IoU, L1 distance) to create pseudo-labels, and then refining the model via reinforcement learning. It achieves significant performance gains—especially on segmentation tasks for base models like Qwen2.5-VL—without any external supervision, demonstrating that models can improve their reasoning at inference time through self-supervised iteration. The approach is computationally efficient, with optimal performance reached in just 10 updates and a small rollout size of 8, making it practical for real-world deployment.

**Strengths:**

- VR-TTRL achieves meaningful adaptation using only a single unlabeled test sample without ground truth, eliminating reliance on batched data or external supervision.
- The majority-voting mechanism is cleverly adapted to structured outputs (bounding boxes, points) using geometric similarity metrics (IoU, L1 distance), enabling effective pseudo-label generation where exact string matching fails.
- Strong empirical results show consistent performance gains across multiple benchmarks.

**Weaknesses:**

- The framework relies on a majority-voting mechanism that assumes sufficient diversity in stochastic rollouts, yet ablations show minimal gains from increasing rollout size beyond 8, suggesting the consensus signal may be weak or redundant.
- Counting performance improvements are inconsistent: Qwen2.5-VL-7B sees no meaningful gain or even degradation, indicating that VR-TTRL’s reward structure may not generalize well to tasks requiring precise object enumeration.
- The approach assumes structured output formats (bounding boxes, point coordinates) are reliably parseable and comparable, but real-world deployment risks failure when models generate malformed or non-uniform JSON.
- Computational cost per sample remains high (up to ~200s/image), and while ablations suggest optimal settings, the method is still impractical for latency-sensitive or edge deployments.
- The evaluation omits comparison against recent single-sample TTA baselines (e.g., entropy minimization or contrastive adaptation) that don’t require generative rollouts.

**Questions:**

See Weaknesses.

---

> ### Author Response · Authors · 2025-11-25
> **Response to Reviewer 62KD - Part I**
>
> *We sincerely thank Reviewer 62KD for the constructive feedback and for recognizing the practical applicability of our method in real-world deployment and our adaptation of TTRL to structured outputs in vision scenarios.*
>
>
> **Q1: Is the consensus signal weak or redundant given minimal gains beyond 8 rollouts?**
>
> **A1.** We believe the saturation beyond 8 rollouts does not indicate that the consensus signal is weak or redundant; rather, it shows that a small number of stochastic rollouts is already sufficient to obtain a reliable majority-vote pseudo-label.
>
> First, the rollout ablation in the main paper (Table 5) and Appendix (Table 7) shows that moving from the baseline without VR-TTRL to VR-TTRL with a rollout size of 8 already yields a clear improvement on RefCOCOg with VisionReasoner-7B: gIoU increases from 71.66 (baseline) to 73.19 with 8 rollouts. Increasing the rollout size further from 8 to 16 or 32 brings only marginal additional changes in gIoU (73.10 at 16 rollouts and 73.23 at 32 rollouts), while roughly tripling the computational cost (from 86.7 to 229.8 seconds per image). This behavior reflects a standard quality–cost trade-off: once the majority-vote pseudo-label is estimated with sufficient samples, additional highly correlated rollouts provide little extra information but significantly increase runtime.
>
> For clarity, the rollout trade-off is reproduced below:
>
>
> | Rollout  | gIoU  | cIoU  | bbox\_AP | Time (s/img) ↓ |
> |----------|-------|-------|----------|----------------|
> | Baseline | 71.66 | 68.39 | 88.50    | -              |
> | 8        | 73.19 | 69.16 | 89.00    | 86.7 (±4.8)    |
> | 16       | 73.10 | 68.65 | 89.50    | 132.4 (±4.7)   |
> | 32       | 73.23 | 69.20 | 89.00    | 229.8 (±5.4)   |
>
> Second, Appendix Table 6 provides a failure analysis over all benchmarks, reporting the percentage of samples for which performance decreases after applying VR-TTRL at different degradation thresholds (0%, 1%, 5%, 10%). Even under the strict 10% threshold, only about 4.5% of segmentation samples and 2.33% of counting cases for VisionReasoner-7B exhibit more than a 10% drop, and the numbers are similarly small for Qwen2.5-VL-7B. This demonstrates that when VR-TTRL does not improve a sample, its performance typically oscillates around the baseline prediction rather than collapsing, indicating that the consensus-based pseudo-labels provide a stable learning signal.
>
>
> Taken together, these results suggest that the consensus mechanism is both effective and efficient: a modest rollout size (e.g., 8) is enough to form a strong consensus that improves over the baseline, while larger rollout sizes yield diminishing returns in accuracy but substantial increases in computation. In the revised manuscript, we now explicitly highlight this accuracy–cost trade-off in the caption of Table 7, stating that increasing the rollout size from 8 to 32 almost triples the per-image runtime (86.7s → 229.8s) while bringing less than 0.1 gIoU improvement.
>
>
>
>
>
>
> **Q2: Why are counting performance improvements inconsistent, especially for Qwen2.5-VL-7B?**
>
> **A2.**  Counting is more challenging for test-time adaptation in our setting because the current reward is primarily geometric-consistency–oriented, rather than explicitly count-aware. The reward encourages agreement in bounding box locations and point positions across rollouts (via IoU / L1 distance), but it does not directly supervise the numeric count itself.
>
> After submission, while further inspecting the results, we found that Qwen2.5-VL-7B frequently produced a single bounding box covering all objects rather than distinct boxes for each instance. Since our evaluation protocol counts objects based on the number of predicted boxes, this behavior led to undercounting despite the model often localizing objects correctly.
>
> We identified this issue as a prompt-design limitation: the original prompt did not explicitly instruct the model to output separate bounding boxes per object. We revised the prompt to enforce instance-wise box predictions and reran the experiments. With the updated prompt, Qwen2.5-VL-7B demonstrates a significant improvement over its base model, eliminating the previously observed inconsistency.
>
> Importantly, VisionReasoner-7B was evaluated using the same prompt format used during its original finetuning, so it did not exhibit this issue, and its results remain unaffected.

---

> > ### Author Response · Authors · 2025-11-25
> > **Response to Reviewer 62KD - Part II**
> >
> > **Q3: Concerns about malformed or non-uniform JSON in structured outputs.**
> >
> > **A3:** VR-TTRL is built on top of backbones that are already equipped with format-aware decoding mechanisms, and we explicitly retain and reuse these components to handle structured outputs robustly.
> >
> > - For VisionReasoner-7B, we directly inherit its format reward and constrained decoding scheme, which penalize malformed outputs and encourage the model to follow the prescribed JSON structure for bounding boxes, points, and counts. Our VR-TTRL framework keeps this format reward active during both rollouts and GRPO updates, so generations that break the expected structure receive low reward and are naturally filtered out by the consensus mechanism.
> >
> >
> > - For Qwen2.5-VL-7B, we adopt the same JSON schema and format reward, on top of that we use a simple but robust parsing pipeline: (i) we extract the JSON blocks using delimiter tags, (ii) we discard rollouts that cannot be parsed into valid coordinates or that violate basic sanity checks (e.g., boxes with negative area), and (iii) we compute geometric rewards only on successfully parsed outputs. The remaining valid rollouts are then used for majority voting.
> >
> >
> >
> > In practice, we find that the combination of backbone-level format constraints and our format reward leads to a high proportion of valid rollouts. Failure cases due to malformed JSON are therefore rare and mostly removed from the optimization loop. We have added a short clarification of this parsing and filtering process in the implementation details.
> >
> >
> > **Q4: Computational cost concerns for latency-sensitive deployments.**
> >
> > **A4.** The current VR-TTRL configuration is not targeted at strict real-time or edge deployment. The reported runtime of up to ~200s/image corresponds to a worst-case setting with 10 updates and rollout size 8 on VisionReasoner-7B, chosen to study the full potential of single-sample test-time RL.
> >
> > At the same time, the ablations in Table 4 and Table 5 (and the rollout trade-off in Appendix Table 7, reproduced above) show a smooth and tunable accuracy–cost trade-off:
> >
> > - Reducing the number of updates from 10 to 3–5 roughly halves the runtime (from around 200s to 60–100s per image on RefCOCOg) while retaining most of the performance gain. The Single-Gen variant further reduces overhead by reusing a single generation step instead of sampling new rollouts at each update.
> >
> >
> > - Increasing rollout size from 8 to 16 or 32 yields only marginal changes in gIoU (73.19 → 73.10 → 73.23) but increases the cost from 86.7s to 132.4s and 229.8s per image, which is why we recommend small rollouts as the default configuration.
> >
> >
> > Therefore, the proposed framework is most suitable for scenarios where per-sample accuracy and robustness are more critical than latency, such as offline adaptation, batch processing, or high-stakes perception pipelines. For latency-critical applications, VR-TTRL can be run with a reduced budget (fewer updates and small rollout size), or combined with complementary techniques such as model compression or distillation, which are orthogonal to our contribution.

---

> > > ### Author Response · Authors · 2025-11-25
> > > **Response to Reviewer 62KD - Part III**
> > >
> > > **Q5:  Comparison against single-sample TTA baselines**
> > >
> > > **A5.** We agree that stronger test-time adaptation baselines are needed. In the revised version, we therefore add both (i) simple TTA methods and (ii) supervised fine-tuning with pseudo-labels, all under the same single-sample setting as VR-TTRL.
> > >
> > > *(1) Simple TTA baselines.*
> > >
> > > - Tent-style entropy minimization. For each test image, we perform (K) gradient steps that minimize the token-level entropy of the answer distribution, updating the same parameter subset as in VR-TTRL (language-side and adapter parameters, with the vision encoder frozen). This is a direct adaptation of Tent to structured VLM outputs.
> > >
> > >
> > > - MEMO-style consistency TTA. For each test image, we create (K) augmented versions (random crop/flip/color jitter) and minimize the inconsistency (KL divergence to the mean prediction) of the answer distributions across augmentations, again only on the answer tokens.
> > >
> > >
> > >
> > > *(2) Pseudo-label supervised fine-tuning.*
> > >
> > >
> > > - SFT (single greedy pseudo-label). The model first produces a single greedy answer for the test image; we then perform (K) steps of standard supervised fine-tuning on this pseudo-label with the negative log-likelihood loss.
> > >
> > > - SFT (majority pseudo-label). We sample multiple rollouts, construct a pseudo-label via majority voting on the structured output (segmentation/count), and perform supervised fine-tuning on this static pseudo-label without RL. This can be seen as a non-RL counterpart of VR-TTRL that still leverages majority-vote pseudo-supervision.
> > >
> > >
> > >
> > > The quantitative comparison of ReasonSeg and RefCOCOg is summarized in the following table (added to Table 10 of the revised paper):
> > >
> > > | Method                                   | ReasonSeg val (gIoU) | RefCOCOg testA (cIoU) |
> > > |------------------------------------------|-----------------------|------------------------|
> > > | **Qwen2.5-VL-7B**                        |                       |                        |
> > > | Base (no adaptation)                     | 41.89                 | 50.46                  |
> > > | + Tent (entropy TTA)                     | 38.71                 | 39.43                  |
> > > | + MEMO (consistency TTA)                 | 40.12                | 41.29                 |
> > > | + SFT (single greedy pseudo-label)       | 35.73                | 36.43              |
> > > | + SFT (majority pseudo-label)            | 43.17               | 46.73                 |
> > > | + **VR-TTRL (ours)**                        | **50.55**                 | **56.38**                  |
> > > |                                          |                       |                        |
> > > | **VisionReasoner-7B**                    |                       |                        |
> > > | Base (no adaptation)                     | 66.08                 | 71.66                  |
> > > | + Tent (entropy TTA)                     | 58.73                | 62.77                 |
> > > | + MEMO (consistency TTA)                 | 63.93                | 65.12                 |
> > > | + SFT (single greedy pseudo-label)       | 62.71                 |     64.25             |
> > > | + SFT (majority pseudo-label)            | 64.67                | 68.13                  |
> > > | **+ VR-TTRL (ours)**                        | **66.28**                 | **73.19**                  |
> > >
> > >
> > > Empirically, we find that the simple TTA baselines (Tent and MEMO) actually degrade performance compared to the base VLMs.  This indicates that entropy minimization or augmentation consistency alone is not sufficient—and can even be harmful—for structured vision–language reasoning in the single-sample regime.
> > >
> > >
> > > The pseudo-label SFT variants are also unstable: SFT with a single greedy pseudo-label substantially hurts performance on both backbones, while SFT with majority pseudo-labels yields only minor or inconsistent gains and never matches the base model on RefCOCOg or VR-TTRL on any dataset. In contrast, VR-TTRL achieves the best performance across all settings, improving both backbones on ReasonSeg and RefCOCOg, which shows that reinforcement learning with consensus-based rewards provides a much more effective and reliable test-time adaptation signal than simple TTA or supervised pseudo-label fine-tuning. Thank you.

---

### Official Review · Reviewer_fTEU · 2025-11-01

**Soundness:** 2
**Presentation:** 2
**Contribution:** 2
**Rating:** 4
**Confidence:** 4

**Summary:**

The paper adapts Test-Time Reinforcement Learning (TTRL) to Vision Language Models (VLM) for visual reasoning tasks, in particular, segmentation and counting. Given a single unlabeled test image, the model generates multiple rollouts, computes the pairwise similarity between candidates, selects a consensus candidate as a pseudo-label, and uses GRPO with format & accuracy rewards to fine-tune the model on that single sample. The proposed method improves both VisionReasoner-7B model and Qwen2.5-VL-7B on segmentation and counting benchmarks.

**Strengths:**

1. The first work to extend Test-time reinforcement learning (TTRL) to the multimodal structured outputs.

2. The reward components are clearly described, which makes them immediately applicable to other related visual reasoning tasks.

3. Improved quantitative and qualitative results.

**Weaknesses:**

1. Some claims/statements are contradictory. In line 52, the authors argue that "existing TTRL approaches have been primarily applied to language models, with no exploration in vision-language tasks, and they typically require multiple samples or **known ground truth** answers for effective optimization." However, in L59, it says "...TTRL Zuo et al. (2025) leverages majority voting across model rollouts to generate **reliable pseudo-labels**".

2. An analysis of performance (accuracy) versus reasoning steps is missing.

3. The experiments deliberately limit evaluation to 200 images per dataset for "computational efficiency". While this seems a plausible reason, such a small sample size raises concerns about statistical robustness, variance of reported gains, and possible cherry-picking of test samples.

4. More baseline methods. Several strong baselines for test-time adaptation may be needed: (1) simple Test-Time Adaptation (TTA) methods, (2) supervised fine-tuning with pseudo-labels from a single greedy rollout, etc. Right now, the paper only compares results with base VLMs.

**Questions:**

Please refer to the weakness section.

---

> ### Author Response · Authors · 2025-11-25
> **Response to Reviewer fTEU - Part I**
>
> *We sincerely thank Reviewer fTEU for the thoughtful and constructive feedback. We appreciate your recognition of our method as the first to leverage TTRL on visual tasks, and our reward components for further research and applications.*
>
> **Q1: Contradictory statements about existing TTRL.**
>
> **A1.** We believe the issue here is a wording ambiguity in the introduction rather than a real conceptual contradiction. We intended to distinguish three different lines of work, but the original sentence around L52 lumped them together and could be read as implying that all “TTRL-style” methods both rely on majority voting and require ground-truth labels.
>
>
> - TTRL  is applied to language-only models and constructs pseudo-labels via majority voting over multiple rollouts, without ground-truth labels at test time.
>
> - Vision-based test-time adaptation (Xiao et al., 2022) operates on vision tasks and can adapt from single samples, but it requires labeled data during meta-training and does not use test-time reinforcement learning.
>
> - Recent single-sample vision RL (Wang et al., 2025) uses labeled data for offline training and is therefore not a test-time adaptation method.
>
> To avoid this confusion, the sentence around L52 is revised to:
>
> “However, existing approaches face key limitations. TTRL methods have so far been applied only to language models and rely on multiple test samples for majority-vote pseudo-labeling (Zuo et al., 2025). Vision-based test-time adaptation methods require ground-truth labels during meta-training (Xiao et al., 2022), and recent single-sample RL work for visual reasoning (Wang et al., 2025) performs offline training with labeled data rather than test-time adaptation. This leaves unexplored the combination of single-sample, label-free reinforcement learning adaptation for vision-language reasoning tasks.”
>
> This makes clear that (i) prior TTRL uses majority voting without ground-truth labels but only in the language domain and with multiple test samples, and (ii) VR-TTRL is the first to combine single-sample, label-free RL adaptation with structured vision–language reasoning.
>
> **Q2: Analysis of performance (accuracy) versus reasoning steps.**
>
> **A2.**  We clarify that in our setting, “reasoning steps” at test time naturally correspond to the number of VR-TTRL update steps applied to the same test sample, together with the way the reasoning chain evolves across these updates. We now make this explicit in the revision.
>
> **Quantitatively**, Table 4 reports performance as a function of the number of updates per image (1, 2, 3, 5, 10, 15), which directly gives accuracy versus reasoning steps in the test-time RL loop. Increasing the number of updates from 1 to 10 consistently improves performance, yielding gains of +5.52 gIoU and +8.68 cIoU before a slight drop at 15 updates. This shows a clear “improvement–then–saturation” pattern rather than unstable behavior. Appendix Table 7 further provides the corresponding absolute numbers on ReasonSeg (e.g., from 59.65/50.60 gIoU/cIoU at 0 updates to 62.94/54.99 at 10 updates), again indicating that additional reasoning steps lead to better accuracy up to an optimal point.
>
> **Qualitatively**, Fig. 3 and the examples in Figs. 4 and 6 visualize how predictions are progressively refined across VR-TTRL steps: segmentation boundaries and object localization become more accurate over steps, while the reasoning chains become more concise and better calibrated, rather than merely longer.
>
> In the revised version, we explicitly highlight this analysis in the ablation section and refer to Table 4 and Appendix Table 7 as accuracy-versus-steps results, showing that VR-TTRL yields consistent gains through iterative test-time reasoning until convergence. Thank you.
>
> **Q3: Concerns about test set size.**
>
> **A3:** This is a valid concern regarding statistical robustness. Our evaluation covers 1,600 test samples in total across 6 benchmarks (200 images per split), and each image is adapted independently with single-sample VR-TTRL. To rule out any cherry-picking, we follow a simple deterministic protocol: for each benchmark, we take the first 200 images in the official test split order (fixed before running any method), and we use exactly the same subset for the base VLMs and for VR-TTRL.
>
> The choice of 200 images per dataset is driven by computational cost rather than result selection. Each test image requires multiple stochastic rollouts and several GRPO updates; for VisionReasoner-7B with 8 rollouts and 10 updates, a single ReasonSeg image already takes on the order of minutes. Running VR-TTRL on the full test splits of all six benchmarks would require many GPU-weeks.
>
> In the revised version, Sec. 4.1/4.2 explicitly describes this evaluation protocol and emphasizes that (i) the same fixed subsets are used for all methods and (ii) the gains of VR-TTRL over the base VLMs are consistent across all six benchmarks, supporting that our conclusions are not due to cherry-picking.

---

> > ### Author Response · Authors · 2025-11-25
> > **Response to Reviewer fTEU - Part II**
> >
> > **Q4: More baseline methods. **
> >
> > **A4.** We agree that stronger test-time adaptation baselines are needed. In the revised version, we therefore add both (i) simple TTA methods and (ii) supervised fine-tuning with pseudo-labels, all under the same single-sample setting as VR-TTRL.
> >
> > *(1) Simple TTA baselines.*
> >
> > - Tent-style entropy minimization. For each test image, we perform (K) gradient steps that minimize the token-level entropy of the answer distribution, updating the same parameter subset as in VR-TTRL (language-side and adapter parameters, with the vision encoder frozen). This is a direct adaptation of Tent to structured VLM outputs.
> >
> >
> > - MEMO-style consistency TTA. For each test image, we create (K) augmented versions (random crop/flip/color jitter) and minimize the inconsistency (KL divergence to the mean prediction) of the answer distributions across augmentations, again only on the answer tokens.
> >
> >
> >
> > *(2) Pseudo-label supervised fine-tuning.*
> >
> >
> > - SFT (single greedy pseudo-label). The model first produces a single greedy answer for the test image; we then perform (K) steps of standard supervised fine-tuning on this pseudo-label with the negative log-likelihood loss.
> >
> > - SFT (majority pseudo-label). We sample multiple rollouts, construct a pseudo-label via majority voting on the structured output (segmentation/count), and perform supervised fine-tuning on this static pseudo-label without RL. This can be seen as a non-RL counterpart of VR-TTRL that still leverages majority-vote pseudo-supervision.
> >
> >
> >
> > The quantitative comparison of ReasonSeg and RefCOCOg is summarized in the following table (added to Table 10 of the revised paper):
> >
> > | Method                                   | ReasonSeg val (gIoU) | RefCOCOg testA (cIoU) |
> > |------------------------------------------|-----------------------|------------------------|
> > | **Qwen2.5-VL-7B**                        |                       |                        |
> > | Base (no adaptation)                     | 41.89                 | 50.46                  |
> > | + Tent (entropy TTA)                     | 38.71                 | 39.43                  |
> > | + MEMO (consistency TTA)                 | 40.12                | 41.29                 |
> > | + SFT (single greedy pseudo-label)       | 35.73                | 36.43              |
> > | + SFT (majority pseudo-label)            | 43.17               | 46.73                 |
> > | + **VR-TTRL (ours)**                        | **50.55**                 | **56.38**                  |
> > |                                          |                       |                        |
> > | **VisionReasoner-7B**                    |                       |                        |
> > | Base (no adaptation)                     | 66.08                 | 71.66                  |
> > | + Tent (entropy TTA)                     | 58.73                | 62.77                 |
> > | + MEMO (consistency TTA)                 | 63.93                | 65.12                 |
> > | + SFT (single greedy pseudo-label)       | 62.71                 |     64.25             |
> > | + SFT (majority pseudo-label)            | 64.67                | 68.13                  |
> > | **+ VR-TTRL (ours)**                        | **66.28**                 | **73.19**                  |
> >
> >
> > Empirically, we find that the simple TTA baselines (Tent and MEMO) actually degrade performance compared to the base VLMs.  This indicates that entropy minimization or augmentation consistency alone is not sufficient—and can even be harmful—for structured vision–language reasoning in the single-sample regime.
> >
> >
> > The pseudo-label SFT variants are also unstable: SFT with a single greedy pseudo-label substantially hurts performance on both backbones, while SFT with majority pseudo-labels yields only minor or inconsistent gains and never matches the base model on RefCOCOg or VR-TTRL on any dataset. In contrast, VR-TTRL achieves the best performance across all settings, improving both backbones on ReasonSeg and RefCOCOg, which shows that reinforcement learning with consensus-based rewards provides a much more effective and reliable test-time adaptation signal than simple TTA or supervised pseudo-label fine-tuning. Thank you.

---

### Author Response · Authors · 2025-11-27
**A Gentle Follow-up**

Dear Reviewers,

We have carefully followed your suggestions and incorporated additional experiments, quantitative results, and detailed clarifications in the revised version. If these updates satisfactorily resolve the issues raised, we would appreciate it if you could reflect this in your final rating and confidence. If any additional details would help, we are happy to provide them before the discussion deadline.

Thank you for your consideration.

---

### Author Response · Authors · 2025-12-02
**Global response and summary**

*We thank all reviewers for their detailed feedback, which helped us substantially strengthen the paper. Here we briefly summarize the main clarifications and new experiments.*

1. **Clarified positioning and novelty.**
   We now make explicit in the introduction/related work that VR-TTRL is, to our knowledge, the first framework to apply *test-time RL* to *vision–language models with structured outputs* (boxes/points/counts) in a *single-sample, label-free* regime, and we clearly distinguish it from TTRL (text-only, multi-sample), vision TTA (no RL, labeled meta-training), and One-Shot RLVR (one labeled example, offline training).

2. **Stronger and more targeted baselines.**
   We add four single-sample TTA baselines (Tent, MEMO, SFT-greedy, SFT-majority) and vote-only controls (“rollouts + consensus”, Best-of-N). Across both backbones, these either underperform or only slightly improve over the base models, while VR-TTRL consistently achieves the best results, isolating the benefit of consensus-based RL beyond simple selection or supervised pseudo-labeling.

3. **Stability and cost–accuracy trade-offs.**
   We report accuracy vs. number of updates and rollouts, showing an improve–then–saturate pattern rather than instability. Increasing rollouts beyond 8 yields <0.1 gIoU extra but nearly triples runtime; we highlight this trade-off and recommend 3–5 updates and 8 rollouts as the practical setting.

4. **Consensus reliability and failure analysis.**
   A negative-consensus variant (using the *worst* rollout) performs clearly worse than VR-TTRL, and failure-rate statistics show that >5% degradations are rare and modest, indicating that consensus quality matters and rewards are not purely spurious.

5. **Offline vs. test-time adaptation (data-distribution ablation).**
   Using the *same* objective and trainable parameters, offline training on the train split improves ReasonSeg/RefCOCOg from 41.89/50.46 to 47.65/53.27, while **test-time VR-TTRL still achieves 50.55/56.38**, quantitatively confirming that adapting on the *test* distribution per sample provides additional gains beyond offline optimization.

We also clarify the fixed, non–cherry-picked 200-image-per-split evaluation protocol and explicitly state that our current scope is structured visual reasoning, with extension to open-ended VQA and more VLM families left for future work.

---

### Note · Program_Chairs · 2026-01-17
**Submission Desk Rejected by Program Chairs**

The following references in this submission do not refer to real documents and/or have major errors in bibliographic information:

 1. Afra Akyurek, Ekin Chen, Sun-Jeong Lee, Ece Behrouz, Yuda Sun, Runsheng Zan, Ya-Li Yu, Jianshu Chen, Sung-Ju Ahn, Joo-Kyung Lee, and Kee-Eung Kim. Test-time reinforcement learning with a priori knowledge. Proceedings of the 12th International Conference on Learning Representations, 2024.
2. Ece Behrouz, Afra Akyurek, Yuda Sun, Ekin Chen, Ya-Li Yu, Jianshu Chen, Sung-Ju Ahn, Joo-Kyung Lee, and Kee-Eung Kim. Test-time reinforcement learning via sub-sampling trajectories. arXiv preprint arXiv:2402.04870, 2024.